# Dealing with Non-Stationarity in MARL via Trust-Region Decomposition

**Wenhao Li, Xiangfeng Wang**[*]**, Bo Jin**[*]**, Junjie Sheng**
School of Computer Science and Technology
East China Normal University
Shanghai, China
{52194501026@stu, xfwang@cs, bjin@cs, 52194501003@stu}.ecnu.edu.cn

**Hongyuan Zha**
School of Data Science, The Chinese University of Hong Kong (Shenzhen)
Shenzhen Institute of Artificial Intelligence and Robotics for Society
Shenzhen, China
zhahy@cuhk.edu.cn

## Abstract

Non-stationarity is one thorny issue in cooperative multi-agent reinforcement learning (MARL). One of the reasons is the policy changes of agents during the learning process. Some existing works have discussed various consequences caused by non-stationarity with several kinds of measurement indicators. This makes the objectives or goals of existing algorithms are inevitably inconsistent and disparate. In this paper, we introduce a novel notion, the $\delta\text{-}stationarity$ measurement, to explicitly measure the non-stationarity of a policy sequence, which can be further proved to be bounded by the KL-divergence of consecutive joint policies. A straightforward but highly non-trivial way is to control the joint policies' divergence, which is difficult to estimate accurately by imposing the trust-region constraint on the joint policy. Although it has lower computational complexity to decompose the joint policy and impose trust-region constraints on the factorized policies, simple policy factorization like mean-field approximation will lead to more considerable policy divergence, which can be considered as the trust-region decomposition dilemma. We model the joint policy as a pairwise Markov random field and propose a trust-region decomposition network (TRD-Net) based on message passing to estimate the joint policy divergence more accurately. The **M**ulti-**A**gent **M**irror descent policy algorithm with **T**rust region decomposition, called **MAMT**, is established by adjusting the trust-region of the local policies adaptively in an end-to-end manner. MAMT can approximately constrain the consecutive joint policies' divergence to satisfy $\delta$-stationarity and alleviate the non-stationarity problem. Our method can bring noticeable and stable performance improvement compared with baselines in cooperative tasks of different complexity.

## 1 Introduction

Learning how to achieve effective collaboration in multi-agent decision-making tasks, such as multi-player games (Berner et al., 2019; Vinyals et al., 2019; Ye et al., 2020), resource allocation (Zimmer et al., 2021; Sheng et al., 2022), and network routing (Mao et al., 2020a;b), is a significant problem in cooperative multi-agent reinforcement learning (MARL). Although deep reinforcement learning (RL) has achieved great success in single-agent environments, its adaptation to the multi-agent system (MAS) still faces many challenges due to the complicated interactions among agents. This paper focuses on one of these thorny issues, i.e., non-stationarity, caused by changing agents' policies during the learning process. Specifically, the state transition function and the reward function of each agent depend on the joint action of all agents. The policy change of other agents leads to the change

---

[*]Corresponding authors: Xiangfeng Wang and Bo Jin

of above two functions for each agent. Recently, many works have been proposed to deal with this non-stationarity problem. These works can be divided into two categories (Papoudakis et al., 2019): targeted modifications on standard RL learning schemes (Lowe et al., 2017; Foerster et al., 2018b; Iqbal & Sha, 2019; Baker et al., 2019), and opponent information estimation and sharing (Raileanu et al., 2018; Foerster et al., 2018a; Rabinowitz et al., 2018; Al-Shedivat et al., 2018).

Both categories aim to mitigate the negative impact of policy changes to solve the non-stationarity problem. These algorithms have studied the various consequences caused by non-stationarity and put forward several indicators to measure these consequences. The objectives or goals of these algorithms are inevitably inconsistent and disparate, which makes them ineffective in general. Additionally, those methods also require either an unexpansive training scheme or excessive information exchange, which significantly increase the training costs. Recently, some work has shown that naive parameter sharing, a particular case of opponent information sharing, can effectively alleviate the non-stationarity problem (Gupta et al., 2017; Terry et al., 2020). However, we can prove that naive parameter sharing can lead to an exponentially worse suboptimal outcome with the increasing number of agents.

From the perspective of each agent, solving the non-stationarity problem in cooperative MARL can be transformed into solving multiple non-stationary Markov decision processes (MDPs) (Jaksch et al., 2010; Ortner et al., 2020; Cheung et al., 2019; Mao et al., 2021). In these MDPs, *the interactive environment is composed of multiple agents*, and the reason for the non-stationary environment is the changing of agent policies. If the policies of all agents change slowly, the environment can remain stable. In this case, each agent could converge to optimality, i.e., pursue the best response to other agents, which will alleviate the non-stationarity problem. Moreover, the changing opponents will significantly hinder the agent's learning in MAS (Radanovic et al., 2019; Lee et al., 2020; Mao et al., 2021). To address this issue, a popular class of RL algorithms focus on limiting the divergence between consecutive policies of the learned policy sequence by imposing direct constraints or regularization terms. These methods are referred to as *trust-region-based* or *proximity-based* algorithms (Schulman et al., 2015; 2017; Tomar et al., 2020). However, the precise connection between joint policy divergence and the non-stationarity is still unclear, which motivates us to analyze the relationship theoretically. Furthermore, directly adding the trust-region constraint to the *joint* policy will make the problem intractable. Therefore, effective factorization of the joint policy and trust-region constraint is the key to improve the algorithm's efficiency (Oliehoek et al., 2008).

In this paper, we propose a novel notion called $\delta$-stationarity to measure the stationarity of a given policy sequence. The core of $\delta$-stationarity is the *opponent switching cost*, which is inspired by the *local switching cost* (Bai et al., 2019; Gao et al., 2021) that measure the changing behavior of an single-agent RL, and is used to measures the changing of the agent's joint behavior. Furthermore, the relationship where $\delta$-stationarity is bounded by the KL-divergence of consecutive joint policies is theoretically established. Similar to existing works (Ortner et al., 2020; Cheung et al., 2019; Mao et al., 2021), which use dynamic regret (Jaksch et al., 2010) to measure the optimality of solving non-stationary MDPs algorithms, we can also prove a $\tilde{O}(D_{\max}^{3/2}|\mathcal{O}|^{1/2}\delta_i^{1/4}T)$ dynamic regret bound for each agent. This provides theoretical support to impose the trust-region constraint on the joint policy to alleviate the non-stationarity problem. However, directly dealing with the trust-region constraint on the joint policy is computationally expensive. The natural idea is to decompose the joint policy while imposing trust-region constraints on the factorized policies. However, simple policy factorization like mean-field approximation will lead to sinificant policy divergence, which is considered as the *trust-region decomposition dilemma*.

To this end, we model the joint policy as a Markov random field and propose a trust-region decomposition network (TRD-Net) to adaptively factorize the trust-region of the joint policy and approximately satisfy the $\delta$-stationarity. Local trust-region constraints are imposed on factorized policies, which can be efficiently solved through mirror descent. The TRD-Net constructs the relationship among factorized policies, factorized trust-regions, and the estimated joint policy divergence. To accurately estimate the joint policy divergence, an auxiliary task, which borrowes the idea in offline RL to measure the degree of out-of-distribution, is constructed to train the TRD-Net. The proposed algorithm is denoted as **M**ulti-**A**gent **M**irror descent policy optimization with **T**rust region decomposition, i.e., **MAMT**. MAMT can alleviate the non-stationarity problem and factorize joint trust-region constraint into trust-region constraints on local policies, which could significantly improve the learning effectiveness and robustness. Our contributions mainly consist of the following: 1) We propose a formal non-stationarity definition of cooperative MARL, $\delta$-stationarity, which is derived from the

local switching cost. $\delta$-stationarity is bounded by the joint policy divergence and bounds the dynamic regret of each agent. 2) The novel trust-region decomposition scheme based on message passing and mirror descent could approximately satisfy the $\delta$-stationarity through a computationally efficient way; 3) Our proposed algorithm could bring noticeable and stable performance improvement in multiple cooperative tasks of different complexity than baselines.

## 2 THE STATIONARITY OF THE LEARNING PROCEDURE

In our work, we consider a cooperative multi-agent task that can be modelled by a cooperative POSG (Hansen et al., 2004) $\langle \mathcal{I}, \mathcal{S}, \{\mathcal{A}_i\}_{i=1}^n, \{\mathcal{O}_i\}_{i=1}^n, \mathcal{P}, \mathcal{E}, \{\mathcal{R}_i\}_{i=1}^n \rangle$, where $\mathcal{I}$ represents the $n$-agent space. $s \in \mathcal{S}$ represents the true state of the environment. We consider partially observable settings, where agent $i$ only accessible to a local observation $o_i \in \mathcal{O}_i$ according to the emission function $\mathcal{E}(o_i|s)$. At each timestep, each agent $i$ selects an action $a_i \in \pi_i(a \mid o_i)$, forming a joint action $\boldsymbol{a} = \langle a_1, \ldots, a_n \rangle \in \times \mathcal{A}_i$, results in the next state $s'$ according to the transition function $P(s' \mid s, \boldsymbol{a})$ and a reward $r_i = \mathcal{R}_i(s, \boldsymbol{a})$ and $\frac{\partial \mathcal{R}_{i'}}{\partial \mathcal{R}_i} \geqslant 0$ where $i$ and $i'$ are a pair of agents. Intuitively, this means that there is no conflict of interest for any pair of agents.

To explore and mitigate the non-stationarity in cooperative MARL, we first need to model the non-stationarity explicitly. To emphasize, the non-stationarity is not an intrinsic attribute of a cooperative POSG but is additionally introduced when using a specific learning algorithm.

As mentioned above, solving the non-stationarity problem in cooperative MARL can be transformed into solving multiple dynamic non-stationary MDPs, where the non-stationary transition probability $\mathcal{T}_i$ and reward function $\mathcal{R}_i$ is caused by the changing of environment (other agents policies). Switching cost is a standard notion in the literature to measure the changing behavior of an single-agent RL algorithm (Cesa-Bianchi et al., 2013; Bai et al., 2019; Gao et al., 2021). We consider the following definition of the (local) switching cost from Bai et al. (2019):

**Definition 1** (Local switching cost). *Let $H$ be the horizon of the MDP and $K$ be the number of episodes that the agent can play. The local switching cost (henceforth also "switching cost") between any pair of policies $(\pi, \pi')$ is defined as the number of $(h, o)$ pairs on which $\pi$ and $\pi'$ are different:*

$$n_{switch}(\pi, \pi') := |\{(h, o) \in [H] \times \mathcal{O} : \pi^h(o) \neq [\pi']^h(o)\}|,$$

*where $[H] := 1, \cdots, H$ and $o, \mathcal{O}$ are the local observation and observation space. For an single-agent RL algorithm that employs policies $(\pi^1, \cdots, \pi^K)$, its local switching cost is defined as $N_{switch} := \sum_{k=1}^{K-1} n_{switch}(\pi_k, \pi_{k+1})$.*

In MARL, it needs more attention that the magnitude of the change of other agents' joint policy at each learning step. Thus, we propose the following *opponent switching cost* to measure the changing behavior of opponents by naturally extending the local switching cost:

**Definition 2** (Opponent switching cost). *Let $H$ be the horizon of the MDP and $K$ be the number of episodes that the agent can play, so that total number of steps $T := HK$. The opponent switching cost of agent $i$ is defined as the maximum Kullback–Leibler divergence of all $(t, \boldsymbol{o})$ pairs between any pair of opponents' joint policies $(\pi_{-i}, \pi'_{-i})$ on which $\pi_{-i}$ and $\pi'_{-i}$ are different:*

$$d_{switch}^i(\pi_{-i}, \pi'_{-i}) := \max\{(t, \boldsymbol{o}) \in [T] \times \mathcal{O} : D_{\mathrm{KL}}(\pi_{-i}^t(\boldsymbol{o}) \| [\pi'_{-i}]^t(\boldsymbol{o})\},$$

*where $[T] := 1, \cdots, T$ and $\boldsymbol{o}, \mathcal{O}$ are the joint observation and observation space.*

The larger the changing magnitude of other agents' joint policy is, the larger the opponent switching cost is, and the more serious non-stationarity problem will be suffered (Radanovic et al., 2019; Lee et al., 2020; Mao et al., 2021). Therefore, opponent switching cost could play an effective role to measure the non-stationarity. Based on the definition of opponent switching cost, the (non-)stationarity of the learning procedure can be further defined as follows:

**Definition 3** ($\delta$-stationarity of the learning procedure). *For a MAS containing $n$ agents, if we have $d_{switch}^i(\pi_{-i}, \pi'_{-i}) \leq \delta_i$, then the learning procedure of agent $i$ is $\delta_i$-stationary. Further, if all agents are $\delta$-stationary with corresponding $\{\delta_i\}$, then the learning procedure of entire multi-agent system is $\delta$-stationary with $\delta = \frac{1}{n} \sum_i \delta_i$.*

There is a strong relationship between the Definition 3 and the changing of the $\mathcal{T}_i, \mathcal{R}_i$ for each agent $i$, which is stated by Lemma 1 and Lemma 2:

**Lemma 1.** *If any agent $i$ satisfies $\delta_i$-stationarity, then the total variation distance of its two consecutive transition probability distribution $p^t(\cdot|\boldsymbol{o}, a_i)$ and $p^{t+1}(\cdot|\boldsymbol{o}, a_i)$ is bounded by $\delta_i$, i.e., $D_{\mathrm{TV}}\left(p^t(\cdot|\boldsymbol{o}, a_i)\|p^{t+1}(\cdot|\boldsymbol{o}, a_i)\right) \leq 2\ln 2 \cdot \delta_i^{1/2}$.*

**Lemma 2.** *Assume the absolute value of joint reward is less than $1$. If any agent $i$ satisfies $\delta_i$-stationarity, then the total variation distance of its two consecutive reward function $r^t(\boldsymbol{o}, a_i)$ and $r^{t+1}(\boldsymbol{o}, a_i)$ is bounded by $\delta_i$, i.e., $D_{\mathrm{TV}}\left(r^t(\boldsymbol{o}, a_i),\ r^{t+1}(\boldsymbol{o}, a_i)\right) \leq 2\ln 2 \cdot \delta_i^{1/2}$.*

Lemma 1 and Lemma 2 indicate that the $\delta$-stationarity is a reasonable mathematical description of the non-stationarity problem (Hernandez-Leal et al., 2017; Papoudakis et al., 2019; Padakandla, 2021), which can measure the non-stationarity of the learning procedure. Besides, we find that controlling the $\delta$-stationarity of the learning procedure can obtain tighter (dynamic) regret bound for each agent $i$, which indicates smaller distance to the set of coarse correlated equilibria (CCE) in finite games (Hannan, 2016; Hart & Mas-Colell, 2000; Hsieh et al., 2021). Based on the Lemma 1 and Lemma 2, Theorem 1 states that the (dynamic) regret of each agent $i$ can be bounded by corresponding $\delta_i$ (the definitions of $D_{\max}$, $B_r$ and $B_p$ are shown in appendix):

**Theorem 1.** *Consider the learning procedure of a MAS satisfies the $\delta$-stationarity and each agent satisfies the $\delta_i$-stationarity. Let $H$ be the horizon and $K$ be the number of episodes, so that total number of steps $T := HK$. In addition, suppose that $T \geq B_r + 2D_{max}B_p > 0$, then a $\tilde{O}(D_{\max}^{3/2}|\mathcal{O}|^{1/2}\delta_i^{1/4}T)$ dynamic regret bound is attained for each agent $i$.*

However, computing the KL-divergence of the consecutive opponents' joint policies $(\pi_{-i}^t, \pi_{-i}^{t+1})$ still intractable. For a $n$-agents system, $n$ constraints on the joint policy divergence should be modeled simultaneously. Before proposing the algorithm, we still need to relax the constraints to increase efficiency. Based on the following theorem, the constraints on the opponents' joint policy divergence can be limited by the joint policy divergence of all agents, independent of the agent number.

**Theorem 2.** *For a multi-agent system, the maximum KL-divergence of all agents' consecutive joint policies $(\pi, \pi')$ is the upper bound of the average opponent switching cost $(1/n) \cdot \sum_{i=1}^{n} d_{switch}^i\left(\pi_{-i}, \pi_{-i}'\right) \leq \max_{\boldsymbol{o}} D_{\mathrm{KL}}\left(\pi(\cdot|\boldsymbol{o})\|\pi'(\cdot|\boldsymbol{o})\right).$*

With the above Theorem 2, it only needs to impose only one trust-region constraint on all agents' joint policy to control the non-stationarity of the entire learning procedure. This allows us to design a MARL algorithm with better effectiveness to approximate the defined $\delta$-stationarity.

## 3 THE PROPOSED MAMT METHOD

According to Theorem 2, we need to properly constrain the maximum divergence of consecutive joint policies to make the learning procedure more stable and efficient. In other words, the algorithm needs to balance between eliminating the non-stationarity (i.e., $\delta \to 0$) and fast learning (i.e., $\delta \to \infty$). To emphasize, based on Definition 3, Theorem 1 and Theorem 2, we can modify the cooperative MARL formulation by adding stationarity constraint. This problem imposes a constraint that the KL divergence is bounded at every point in the state space. While it is motivated by the theory, this problem is impractical to solve due to the large number of constraints. Instead, we can use a heuristic approximation, similar as Schulman et al. (2015), which considers the average KL divergence:

$$\boldsymbol{\pi}^{k+1} \in \arg\max_{\boldsymbol{\pi}} \sum_{i=1}^{n} V_i^{\boldsymbol{\pi}}(\boldsymbol{o}), \forall \boldsymbol{o},\ \text{s.t. } \mathbb{E}_{\boldsymbol{o}\sim\mathcal{D}}\left[\mathbb{E}_{\boldsymbol{a}\sim\boldsymbol{\pi}}\left[D_{\mathrm{KL}}\left(\pi(\boldsymbol{a}\mid\boldsymbol{o}), \pi^k(\boldsymbol{a}\mid\boldsymbol{o})\right)\right]\right] \leq \delta, \quad (1)$$

where $\boldsymbol{\pi}$ represents the joint policy; $\boldsymbol{o} := (o_1, \cdots, o_i, \cdots, o_n)$ represents the joint observation with $n$ be the agent number. $\mathcal{D}$ is the replay buffer. Considering the low sample efficiency of MARL, we did not adapt the fully on-policy learning, but use off-policy training with a small replay buffer. In this way, it can be ensured that samples are less different from the current policy thereby alleviating the instability caused by off-policy training. Directly imposing a constraint on the joint policy as in (1) will make the problem intractable. A straightforward way is to evenly distribute the trust-region to all agents based on mean-field policy approximation (as shown in Figure 6 in the appendix). However, this will bring up the problem which we call *trust-region decomposition dilemma*.

### 3.1 TRUST-REGION DECOMPOSITION DILEMMA

Formally, inspired by Lowe et al. (2017), Foerster et al. (2018b) and Iqbal & Sha (2019), we first introduce the *mean-field approximation* assumption.

**Assumption 1** (Mean-Field Approximation). *We use the mean-field variation family $\pi$ to estimate the joint policy of all agents, i.e., $\pi(\boldsymbol{a}|\boldsymbol{o}) = \prod_{i=1}^{n} \pi_i(a_i|o_i)$.*

Based on mean-field approximation assumption, the joint policy trust-region constraint can be factorized into trust-region constraints of local policies with the following theorem.

**Theorem 3.** *For any consecutive policies which are belong to mean-field variational family $\pi$ and $\pi^k$ in the policy sequence obtained by any learning algorithm, the KL divergence constraint on joint policy in (1) is equivalent with the following summation constraints of local policies, i.e.,*

$$\sum_{i=1}^{n} \mathbb{E}_{o_i \sim \mathcal{D}} \left[ \mathbb{E}_{a_i \sim \pi_i} \left[ D_{\mathrm{KL}}(\pi_i(a_i|o_i), \pi_i^k(a_i|o_i)) \right] \right] \leq \delta. \tag{2}$$

*Further the joint summation trust-region constraint (2) can be equivalently decomposed into the following local trust-region constraints, i.e.,*

$$\mathbb{E}_{o_i \sim \mathcal{D}} \left[ \mathbb{E}_{a_i \sim \pi_i} \left[ D_{\mathrm{KL}}(\pi_i(a_i|o_i), \pi_i^k(a_i|o_i)) \right] \right] \leq \delta_i, \quad \forall i, \tag{3}$$

*with $\sum_{i=1}^{n} \delta_i = \delta$, $0 \leq \delta_i \leq \delta$; $\pi_i$ and $\pi_i^k$ represent the consecutive local policies of agent $i$; $o_i$ and $u_i$ represent the local observation and initial local observation distribution of agent $i$ respectively.*

Due to the summation constraint term (2), it is still need to jointly train the policies of all agents with limited effectiveness and solvability. Therefore, we further equivalently transform (2) into (3) [*]. Based on Theorem 3, problem (1) can be reformulated as

$$\pi^{k+1} \in \arg\max \sum_{i=1}^{n} V_i^{\pi}(\boldsymbol{o}), \forall \boldsymbol{o}, \text{s.t. } \mathbb{E}_{o_i \sim \mathcal{D}} \left[ \mathbb{E}_{a_i \sim \pi_i} \left[ \mathrm{KL}\left( \pi_i\left(a_i \mid o_i\right), \pi_i^k\left(a_i \mid o_i\right) \right) \right] \right] \leq \delta/n, \forall i. \tag{4}$$

The trust-region decomposition scheme adopted in the above, we called MAMD, is based on the mean-field approximation assumption (i.e., formulation (4)). The similar optimization problem is also obtained by Li & He (2020) that try to implement TRPO for MARL through distributed consensus optimization. However, decomposing the trust-region inappropriately will make the algorithm converge to sub-optimal solutions, which can be numerically proved through the following example. By considering a simple MAS with three agents $i, j, k$, we assume that the agent $k$ is independent with two agents $i$ and $j$. Then for the multi-agent system, we have $p_i(\boldsymbol{o}'|\boldsymbol{o}, a_i, a_j, a_k) = p_i(\boldsymbol{o}'|\boldsymbol{o}, a_i, a_j), p_j(\boldsymbol{o}'|\boldsymbol{o}, a_j, a_i, a_k) = p_j(\boldsymbol{o}'|\boldsymbol{o}, a_j, a_i), p_k(\boldsymbol{o}'|\boldsymbol{o}, a_k, a_i, a_j) = p(\boldsymbol{o}'|\boldsymbol{o}, a_k)$.

The change of $k$'s policy cannot affect the state transition probability of $i, j$, and vice versa. As a result, the constraints on the $i, j$ are "insufficient" but the constraint for $k$ is "excessive", when $\delta$ in (2) is decomposed into three parts equally, which is called *trust-region decomposition dilemma*[†] in this paper. Theoretically, the reason for the trust-region decomposition dilemma might be the inaccurate estimation of the joint policy divergence based on the mean-field approximation assumption. The simple summation of local policies' divergence is not equal to the joint policy divergence, while the gap might be enormous. Once the latter cannot be accurately estimated, the non-stationarity cannot be judged accordingly, and imposing constraints on it may run counter to our goals.

## 3.2 THE MAMT ALGORITHM FRAMEWORK

Solving the trust-region decomposition dilemma need to accurately model the relationship between the local policy divergences and the joint policy divergence. However, calculating the joint policy divergence is intractable. MAMT solves this dilemma from another perspective, i.e., *directly learning the relationship* through the following three steps. First, approximating the joint policy divergence; Then, using a differentiable function, i.e., the trust-region decomposition network (TRD-Net), to fit the relationship between the approximated joint policy divergence and local trust-regions; Finally, adaptively adjust local trust regions by optimizing the approximated joint policy divergence. The rest of this section will be organized in three aspects: joint policy divergence approximation, the trust-region decomposition network, and local trust-region optimization.

**Joint Policy Divergence Approximation.** From the analysis of the trust-region decomposition dilemma, it can be seen that the dependence between agents is an essential factor affecting the joint policy divergence. Formally, we use a non-negative real number, *coordination coefficient*, to represent the dependency between two agents[‡]. In the learning procedure, the coordination relationship between

---

[*]In this case, the number of constraints will increase to $n$, however the number of constraints will become $n^2$ if the decomposition technique is imposed on the $n$ joint policy of the other agents.

[†]We verify the existence of trust-region decomposition dilemma through a typical example in the appendix.

[‡]In this paper, we only model the pairwise relationship.

agents is changing with the learning of policies and completing tasks. Therefore, we model the cooperative relationship between two agents based on counterfactual (Foerster et al., 2018b; Jaques et al., 2019). In Jaques et al. (2019), the causal influence reward of agent $i$ w.r.t. opponent $j$ is calculated by $c_{i,j}^t = D_{KL}\left[p\left(a_j^t \mid a_i^t, o_j^t\right) \| p\left(a_j^t \mid o_j^t\right)\right]$, where $p\left(a_j^t \mid s_j^t\right) = \sum_{\tilde{a}_i^t} p\left(a_j^t \mid \tilde{a}_i^t, o_i^t\right) p\left(\tilde{a}_i^t \mid o_i^t\right)$. It can be seen that the greater the causal influence reward, the tighter the coordination between the two agents. However, when calculating the causal influence reward, the agent's policy needs to be modified. That is, the agent needs to explicitly rely on the actions of other agents when making decisions. To solve this problem, we modified the process of calculating the counterfactual baseline in Foerster et al. (2018b) based on the idea of Jaques et al. (2019), and obtained a new way of calculating the coordination coefficient

$$\hat{\mathcal{C}}_{i,j} = \text{softmax}\left(\left|Q_i(\mathbf{a}_{\backslash j}) - Q_i(\mathbf{a})\right|\right), \ \mathcal{C}_{i,j} = \mathbb{1}[\hat{\mathcal{C}}_{i,j} \geq \sigma] \cdot \hat{\mathcal{C}}_{i,j}, \tag{5}$$

where $Q_i(\cdot)$ is the centralized critic of agent $i$. $\mathbf{a}$ is the joint action of all agents, and $\mathbf{a}_{\backslash j}$ is the joint action of all agents expect for $j$; $\sigma$ is the threshold to keep sparsity.

According to Theorem 2, Definition 2 and Assumption 1, after modeling the coordination coefficient between agents, does it mean that the weighted summation of the local policy divergence, i.e., $1/n \cdot \sum_i \mathcal{C}_{i,j} \sum_{j \neq i} \text{KL}[\pi_{\psi_j}'(o_j^t) \| \pi_{\psi_j}(o_j^t)]$ can accurately estimate the joint policy divergence? It can be seen from Definition 3 that the agents' policies do not directly cause the non-stationarity. The agent cannot directly observe the opponent's policy but only the action sequences. This explicit information directly affects the learning process of the agent and leads to non-stationarity. Therefore, we borrowed the idea in offline RL to measure the degree of out-of-distribution through the model-based RL (Kidambi et al., 2020; Yu et al., 2020). Specifically, each agent $i$ has a prediction model $h_{\phi_i^j}$ for each other agent $j$, predicting the other agent's actions based on its local history observation. The prediction model can naturally replace the old policy $\pi'$ since it is trained using historical information. Thus the summation of the divergence between the predicted action distribution and the actual action distribution of all other agents could represent the non-stationarity of the agent $i$, i.e.,

$$\mathcal{D}_{i,\text{ns}}^t = \Pi_{\text{ns}}(\sum_{j \neq i} \mathcal{C}_{i,j}^t(\text{KL}[h_{\phi_i^j}(o_i^t) \| \pi_{\psi_j}(o_j^t)])), \tag{6}$$

where "ns" denotes the "non-stationarity"; $\pi_{\psi_j}$ is the policy of agent $j$ which is parameterized by $\psi_j$; $\Pi_{\text{ns}}$ represents the projection function, which constrains $\mathcal{D}_{i,\text{ns}}^t$ to a specific range. Kim et al. (2020) also introduces inter-agent action prediction, but Kim et al. (2020) is to promote collaboration, and this paper is to estimate the non-stationarity of the joint policy better. The joint policy divergence can then be approximated by the summation of all local non-stationarities $\mathcal{D}_{\text{ns}}^t = \sum_i \mathcal{D}_{i,\text{ns}}^t$.

**Trust-Region Decomposition Network.** To achieve more reasonable joint policy decomposition, recent works (Böhmer et al., 2020; Qu et al., 2020; Li et al., 2021) modeled the joint policy as a Markov random field with pairwise interactions (shown in the Figure 6 in the appendix) based on graph neural networks (GNN). The joint policy divergence is related to the local policies and local trust-regions. Inspired by these methods, a similar mechanism is utilized in this paper to decompose the joint policy divergence into local policies and local trust regions (which can be input to the GNN) with pairwise interactions. The employed GNN is denoted as *trust-region decomposition network* and the network structure is shown in Figure 1. The approximated joint policy divergence $\mathcal{D}_{\text{ns}}$ can then be used as an surrogate supervision signal to train the trust-region decomposition network. Formally, the loss function to learn the trust-region decomposition network can be formulated as

$$\min_\theta \mathcal{L}_{\text{ns}} = (\sum_i \hat{\text{KL}}_i^t - \sum_i \mathcal{D}_{i,\text{ns}}^t)^2, \tag{7}$$

where $\theta$ and $\hat{\text{KL}}_i^t$ are the parameters and the output of the trust-region decomposition network.

**Local Trust-Region Optimization.** Based on the approximated joint policy divergence and the trust-region decomposition network, the learning of local trust-region $\delta_i$ can be formulated by the trade-off between two parts, i.e., the non-stationarity of the learning procedure and the performance of all agents. Formally, the learning objective of $\delta_i$ is to maximize

$$\mathcal{F}(\boldsymbol{\delta}) := \mathbb{E}_{\boldsymbol{o} \sim \mu}[\sum_{i=1}^n V_i^{\boldsymbol{\pi(\delta)}}(\boldsymbol{o}) - \hat{\text{KL}}_i^{\boldsymbol{\pi(\delta)}}(\boldsymbol{o}, \boldsymbol{\delta}; \theta)],$$

where $\boldsymbol{\delta} = \{\delta_i\}_{i=1}^n$ and $\boldsymbol{\pi(\delta)}$ represent the joint policy is related to the $\delta_i$ of all agents; $\hat{\text{KL}}_i^{\boldsymbol{\pi(\delta)}}$ denotes the output of the TRD-Net which is parameterized by $\theta$. Finally, the learning of $\{\delta_i\}$ and $\theta$ can be modeled as a bilevel optimization problem (Dempe & Zemkoho, 2020)

$$\boldsymbol{\delta}^\star = \arg\max \ \mathcal{F}(\boldsymbol{\delta}, \theta^\star(\boldsymbol{\delta})), \quad \text{s.t.} \ \theta^\star(\boldsymbol{\delta}) = \arg\min \ \mathcal{L}_{\text{ns}}(\boldsymbol{\delta}, \theta).$$

Figure 1: The trust-region decomposition network. "MP" denotes message passing. Each agent $i$ first encodes $o_i^t$ and $a_i^t$ and concatenates them with the local trust-region embedding , and gets the agent's embedding (Input Layer). Next, we construct agents as a weighted undirected graph, $C_{*,*}^t$ are calculated in advance, and the updated agent's embedding is obtained through the GNN Layers. Each agent then obtains another agent's embedding. Finally, we concatenate the two embeddings together and input them into KL-Encoder to estimate the current joint policy divergence (Prediction Layer).

We can employ the efficient two-timescale gradient descent method to simultaneously perform gradient update for both $\boldsymbol{\delta}$ and $\theta$. Specifically, we have

$$\boldsymbol{\delta}_{k+1} \leftarrow \boldsymbol{\delta}_k - \alpha_k \cdot \nabla_{\boldsymbol{\delta}} \mathcal{F}(\boldsymbol{\delta}_k, \theta_k), \; \theta_{k+1} \quad \leftarrow \theta_k - \beta_k \cdot \nabla_\theta \mathcal{L}_{\text{ns}}(\boldsymbol{\delta}_k, \theta_k), \; s.t. \; \alpha_k / \beta_k \to 0, \quad (8)$$

where $\frac{\alpha_k}{\beta_k} \to 0$ indicates that $\{\theta_k\}_{k \geq 0}$ updates faster than $\{\boldsymbol{\delta}_k\}_{k \geq 0}$. In practical, we make $\alpha_k$ and $\beta_k$ equal but perform more gradient descent steps on $\theta$, similar as Fujimoto et al. (2018). At the same time, in order to ensure that the updated $\boldsymbol{\delta}_{k+1}$ can meet the constraint, that is, $\sum_i \delta_{i,k+1} < \delta$, we added an additional regular term $\text{ReLU}(\sum_i \delta_{i,k+1} - \delta)$ to $\mathcal{F}(\boldsymbol{\delta})$ during algorithm training.

**Algorithm Summary.** By directly combing the MAAC (Iqbal & Sha, 2019) and the mirror descent technique in MDPO (Tomar et al., 2020), we can get the algorithm framework for solving problem formulated in Equation (1) after decomposing the trust-region constraint. Each agent $i$ has its local policy network $\pi_{\psi_i}$ and a local critic network $Q_{\zeta_i}$, which is similar to MADDPG[§] (Lowe et al., 2017). All critics are centralized updated iteratively by minimizing a joint regression loss function, i.e.,

$$\mathcal{L}_{critic}(\zeta_1, \cdots, \zeta_n) = \sum_{i=1}^n \mathbb{E}_{(o,a,r,o') \sim \mathcal{D}}[(Q_{\zeta_i}(o,a) - y_i)^2], \quad (9)$$

with $y_i = r_i + \gamma \mathbb{E}_{a' \sim \pi_{\bar{\psi}}(o')}[Q_{\bar{\zeta}_i}(o', a') - \alpha \log(\pi_{\psi_i}(a_i' \mid o_i'))]$, and the parameters of the attention module are shared among agents; $\pi_{\bar{\psi}}$ represents the joint target policy (similar as target $Q$-network in DQN) of all agents; $Q_{\bar{\zeta}_i}$ represents the target local critic of agent $i$. $\alpha$ denotes the temperature balancing parameter between maximal entropy and rewards.

As for the individual policy updating step, the mirror descent (Tomar et al., 2020) is introduced to deal with the decomposed local trust-region constraints:

$$\nabla_{\psi_i} J(\pi_\psi) = \mathbb{E}_{o \sim D, a \sim \pi} \left[ \nabla_{\psi_i} \log(\pi_{\psi_i}(a_i \mid o_i)) \left( \text{KL}[\psi_i \| \psi_i^{\text{old}}] / \delta_i + Q_{\zeta_i}(o,a) - b(o, a_{\backslash i}) \right) \right],$$

where $b(o, a_{\backslash i})$ denotes the counterfactual baseline (Foerster et al., 2018b); $\pi_{\psi_i^{\text{old}}}$ represents the policy of agent $i$ at last step; $\delta_i$ is the assigned trust-region range; $\text{KL}[\psi_i \| \psi_i^{\text{old}}] = \log(\pi_{\psi_i}(a_i \mid o_i)) - \log(\pi_{\psi_i^{\text{old}}}(a_i \mid o_i))$. However, if we only perform a single-step SGD, the resulting gradient would be equivalent to vanilla policy gradient and misses the purpose of enforcing the trust-region constraint. As a result, the policy update at each iteration $k$ involves $m$ SGD steps as

$$\psi_{i,k}^{(0)} = \psi_{i,k}, \quad j = 0, \ldots, m-1; \quad \psi_{i,k}^{(j+1)} \leftarrow \psi_{i,k}^{(j)} + \eta \nabla_{\psi_i} J(\pi_\psi)|_{\psi_i = \psi_{i,k}^{(j)}}, \quad \psi_{i,k+1} = \psi_{i,k}^{(m)}.$$

For off-policy mirror descent policy optimization, performing multiple SGD steps at each iteration becomes increasingly time-consuming as the value of $m$ grows, Therefore, similar with Tomar et al. (2020), we resort to staying close to an $m$ step old copy of the current policy while performing a single gradient update at each iteration of the algorithm. This copy is updated every $m$ iterations with the parameters of the current policy. The pseudo-code of MAMT is shown in the Algorithm 1.

## 4 EXPERIMENTS

This section aims to verify the effectiveness of the trust-region constraints, the existence of the trust-region decomposition dilemma, and the capacity of the TRD-Net with 4 cooperative tasks *Spread*, *Multi-Walker*, *Rover-Tower*, *Pursuit* (more details are in the appendix).

---

[§]Our algorithm also follows the centralized critics and decentralized actors framework.

**Baselines**. The fisrt CCDA algorithm **MAD-DPG** (Lowe et al., 2017) is chosen. Our proposed trust-region technique motivates us to set the combination of PPO and MADDPG as another baseline, **MA-PPO** (Yu et al., 2021). The **MAAC** (Iqbal & Sha, 2019) which is based on the attention mechanism is also considered. MA-PPO's policy network also uses an attention mechanism similar to MAAC. **LOLA**[¶] (Foerster et al., 2018a) algorithm that models opponent agents to solve the non-stationarity problem is compared as the baseline. LOLA is only compared in the Spread environment due to the poor scalability. **MAMD**, i.e., MAMT without the TRD-Net, is also considered.

---

**Algorithm 1** MAMT

**Require:** Randomly initialize $\pi_{\psi_i}, Q_{\zeta_i}, \theta$ and $\delta_i^0$
1: Initialize empty replay buffer $\mathcal{D}$
2: Assign $Q_{\bar{\zeta}_i} \leftarrow Q_{\zeta_i}, \{\pi_{\bar{\psi}_i}, \pi_{\psi_i^{\text{old}}}\} \leftarrow \pi_{\psi_i}$
3: **for** i = 1, 2, ... **do**
4:     Collect interaction $\tau_i$ with $\pi_{\psi_i}$
5:     Update replay buffer $\mathcal{D} \leftarrow \{\tau_i\}$
6:     **for** j = 1, 2, ... **do**
7:         Sample transitions $\mathcal{B}$ from $\mathcal{D}$
8:         Compute $\{\mathcal{C}_{i,\backslash i}^t, \mathcal{D}_{i,\text{ns}}^t\}$ with Eq. 5, 6
9:         Update $\delta_i^t, \theta$ with Eq. 8
10:        Update $Q_{\zeta_i}, \pi_{\psi_i}$ by Eq. 9, 10
11:        Update target networks $Q_{\bar{\zeta}_i}$ and $\pi_{\bar{\psi}_i}$
12:        Update old policy $\pi_{\psi_i^{\text{old}}}$

---

**Comparisons**. we compare MAMT with all baselines and get three conclusions:

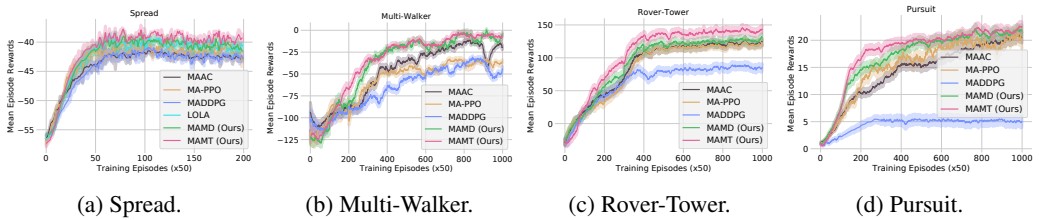

(a) Spread.      (b) Multi-Walker.      (c) Rover-Tower.      (d) Pursuit.

Figure 2: The performance comparisons under coordination tasks with different complexity.

*1). Combine the single-agent trust-region technique with MARL directly cannot bring stable performance improvement.* The averaged episode rewards of all methods in four environments are shown in Figure 2. Directly combining the single-agent trust-region technique with MARL cannot bring stable performance improvement by comparing MAAC and MA-PPO. In the *Rover-Tower* and *Pursuit*, MA-PPO did not bring noticeable performance improvement; MA-PPO only has a subtle improvement in the simple *Spread* and even causes performance degradation in *Multi-Walker*. In contrast, MAMT can bring noticeable and stable performance improvements in all environments. It is worth noting that MAMD also shows promising results. This shows that even a simple trust-region decomposition can bring noticeable improvement in simple scenarios. This further verifies the rationality and effectiveness of non-stationarity modeling.

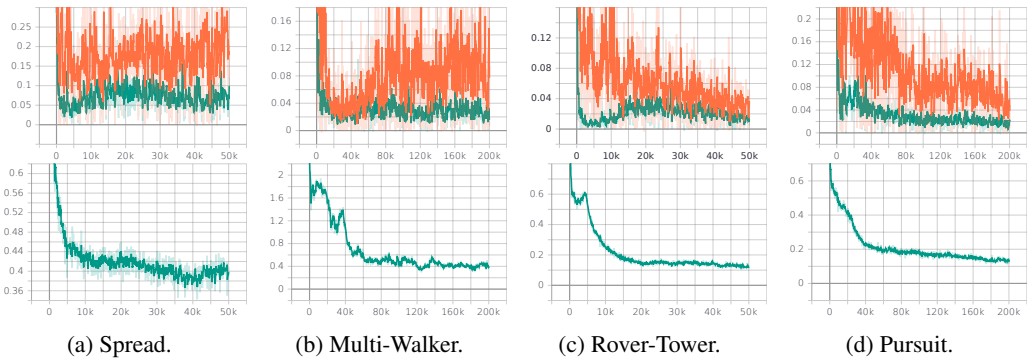

(a) Spread.      (b) Multi-Walker.      (c) Rover-Tower.      (d) Pursuit.

Figure 3: **Upper:** The averaged KL divergence of each agent with the orange and green line representing MAAC and MAMT respectively; **Bottom:** The $\mathcal{D}_{\text{ns}}$ of all agents.

*2). MAMT achieves the trust-region constraint on the local policies and minimizes the non-stationarity of the learning procedure.* Below we conduct an in-depth analysis of why MAMT can noticeably and stably improve the final performance. MAMT uses an end-to-end approach to adaptively adjust the sizes of the local trust-regions according to the current dependencies between agents to constrain

---

[¶]We combine LOLA with DQN to solve more complex stochastic game.

the KL divergence more accurately. Therefore, we analyze MAMT from the two perspectives: the trust-region constraint satisfaction and the non-stationarity satisfaction via surrogate measure. Figure 3 shows the average value of the local KL divergences in different environments. It can be seen from the figure that MAMT controls the policy divergence. The non-stationarity of the learning procedure is challenging to calculate according to the definition, so we use the surrogate measure $\mathcal{D}_{\mathrm{ns}}$ to represents the non-stationarity. It can be seen from Figure 3 that as the learning progresses, the non-stationarity is gradually decreasing.

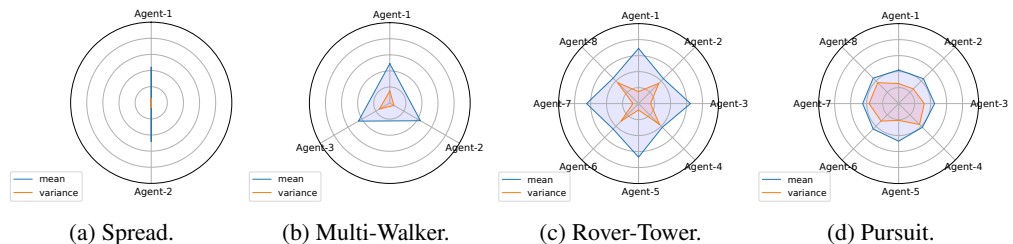

(a) Spread.      (b) Multi-Walker.      (c) Rover-Tower.      (d) Pursuit.

Figure 4: Mean and variance of local trust-region of each agent. The y-axis range from 0.01 to 100.0.

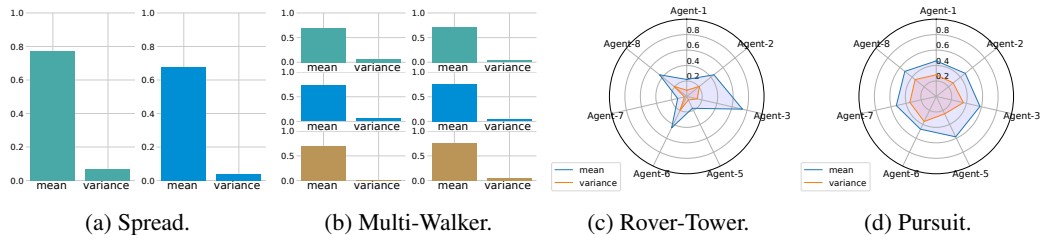

(a) Spread.      (b) Multi-Walker.      (c) Rover-Tower.      (d) Pursuit.

Figure 5: Mean and variance of pairwise coordination coefficients. Different colors represent different agents in (a) and (b). In addition, since there are fewer agents in *Spread* and *Multi-Walker*, the display effect of the radar chart is poor, so we use the histogram for a more precise display in (a) and (b).

*3). Trust-region decomposition works better when there are more agents and more complex coordination relationships.* Figure 2 shows that MAMT and MAMD have similar performance in the *Spread* and *Multi-Walker*, but MAMT is noticeably better than MAMD in the *Rover-Tower* and *Pursuit*. To analyze we separately record the coordination coefficients and the sizes of the trust-regions of the local policies, see Figure 5 and Figure 4. It can be seen that the *Spread* and *Multi-Walker* are relatively simple, and the number of agents is small. This makes the interdependence between agents is very tight (larger mean) and will not change over time (variance is microscopic). Moreover, the corresponding trust-region sizes are similar to coordination coefficients. These figures show that in simple coordination tasks trust-region decomposition is not a severe problem, which leads to the fact that the MAMT cannot bring a noticeable performance improvement. In the other two environments, the dependence between agents is not static (the variance in Figure 5 is more considerable) due to the more significant number of agents and more complex tasks. This also makes the sizes of the trust-regions of the agent's local policies fluctuate significantly in different stages (the variance in Figure 4 is more considerable). MAMT alleviates the trust-region decomposition dilemma through the TRD-Net, which brings a noticeable and stable performance improvement.

## 5 CONCLUSION

In this paper, we define the $\delta$-stationarity to explicitly model the stationarity in the learning procedure of the multi-agent system and provide a theoretical basis for proposing an achievable learning algorithm based on joint policy trust-region constraints. To solve the trust-region decomposition dilemma caused by the mean-field approximation and to estimate the joint policy divergence more accurately, we propose an efficient and robust algorithm MAMT combining message passing and mirror descent with the purpose to satisfy $\delta$-stationarity. Experiments show that MAMT can bring noticeable and stable performance improvement and more suitable for large-scale scenarios with complex coordination relationships between agents.

**Acknowledgment.** This work was supported in part by the National Key Research and Development Program of China (No. 2020AAA0107400), NSFC (No. 12071145), STCSM (No. 19ZR141420, No. 20DZ1100304 and 20DZ1100300), Shanghai Trusted Industry Internet Software Collaborative Innovation Center, and the Fundamental Research Funds for the Central Universities.

**Ethics Statement.** Our method is not a generative model, nor does it involve the training of super-large-scale models. The training data is sampled in the simulated environments, so it does not involve fairness issues. Our method also does not involve model or data stealing and adversarial attacks.

**Reproducibility Statement.** The source code of this paper is available at https://anonymous.4open.science/r/MAMT. We specify all the training details (e.g., hyperparameters, how they were chosen), the error bars (e.g., with respect to the random seed after running experiments multiple times), and the total amount of compute and the type of resource used (e.g., type of GPUs and CPUs) in the Section 4 or the Appendix G. We also cite the creators of partial code of our method and mention the license of the assets in the Appendix G.

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

# Supplementary Material

## Table of Contents

## A  PRELIMINARIES

**Cooperative POSG.** POSG (Hansen et al., 2004) is denoted as a seven-tuple via the stochastic game (or Markov game)

$$\langle \mathcal{I}, \mathcal{S}, \{\mathcal{A}_i\}_{i=1}^n, \{\mathcal{O}_i\}_{i=1}^n, \mathcal{P}, \mathcal{E}, \{\mathcal{R}_i\}_{i=1}^n \rangle,$$

where $n$ denotes the number of agents; $\mathcal{I}$ represents the agent space; $\mathcal{S}$ represents the finite set of states; $\mathcal{A}_i, \mathcal{O}_i$ denote a finite action set and a finite observation set of agent $i \in \mathcal{I}$ respectively; $\mathcal{A} = \mathcal{A}_1 \times \mathcal{A}_2 \times \cdots \times \mathcal{A}_n$ is the finite set of joint actions; $\mathcal{P}(s'|s, \boldsymbol{a})$ denotes the Markovian state transition probability function, where $s, s' \in \mathcal{S}$ represent states of environment and $\boldsymbol{a} = \{a_i\}_{i=1}^n, a_i \in \mathcal{A}_i$ represents the action of agent $i$; $\mathcal{O} = \mathcal{O}_1 \times \mathcal{O}_2 \times \cdots \times \mathcal{O}_n$ is the finite set of joint observations; $\mathcal{E}(\boldsymbol{o}|s)$ is the Markovian observation emission probability function, where $\boldsymbol{o} = \{o_i\}_{i=1}^n, o_i \in \mathcal{O}_i$ represents the local observation of agent $i$; $\mathcal{R}_i : \mathcal{S} \times \mathcal{A} \times \mathcal{S} \to \mathcal{R}$ denotes the reward function of agent $i$ and $r_i \in \mathcal{R}_i$ is the reward of agent $i$. The game in POSG unfolds over a finite or infinite sequence of stages (or timesteps), where the number of stages is called *horizon*. In this paper, we consider the finite horizon case. The objective for each agent is to maximize the expected cumulative reward received during the game. For a cooperative POSG, we quote the definition in Song et al. (2020),

$$\forall i \in \mathcal{I}, \forall i' \in \mathcal{I} \backslash \{i\}, \forall \pi_i \in \Pi_i, \forall \pi_{i'} \in \Pi_{i'}, \frac{\partial \mathcal{R}_{i'}}{\partial \mathcal{R}_i} \geqslant 0,$$

where $i$ and $i'$ are a pair of agents in agent space $\mathcal{I}$; $\pi_i$ and $\pi_{i'}$ are the corresponding policies in the policy space $\Pi_i$ and $\Pi_{i'}$ respectively. Intuitively, this definition means that there is no conflict of interest for any pair of agents.

**Mirror descent method in RL.** The mirror descent method (Beck & Teboulle, 2003) is a typical first-order optimization method, which can be considered an extension of the classical proximal gradient method. In order to minimize the objective function $f(\mathbf{x})$ under a constraint set $\mathbf{x} \in \mathcal{C} \subseteq \mathbb{R}^n$, the basic iterative scheme at iteration $k$+1 can be written as

$$\mathbf{x}^{k+1} \in \underset{\mathbf{x} \in \mathcal{C}}{\arg\min} \left\langle \nabla f\left(\mathbf{x}^k\right), \mathbf{x} - \mathbf{x}^k \right\rangle + \gamma^k B_\psi\left(\mathbf{x}, \mathbf{x}^k\right), \tag{10}$$

where $B_\psi(\mathbf{x}, \mathbf{y}) := \psi(\mathbf{x}) - \psi(\mathbf{x}) - \langle \nabla \psi(\mathbf{y}), \mathbf{x} - \mathbf{y} \rangle$ denotes the Bregman divergence associated with a strongly convex function $\phi$ and $\gamma^k$ is the step size (or learning rate). Each reinforcement learning problem can be formulated as optimization problems from two distinct perspectives, i.e.,

$$\pi^*(\cdot \mid s) \in \underset{\pi}{\arg\max} V^\pi(s), \quad \forall s \in \mathcal{S}; \tag{11a}$$

$$\pi^* \in \underset{\pi}{\arg\max} \mathbb{E}_{s\sim\mu}\left[V^\pi(s)\right]. \tag{11b}$$

Geist et al. (2019) and Shani et al. (2020) have utilized the mirror descent scheme (10) and update the policy iteratively as follows

$$\pi^{k+1}(\cdot|s) \leftarrow \underset{\pi}{\arg\max} \mathbb{E}_{a\sim\pi}\left[A^{\pi^k}(s,a)\right] - \gamma^k \operatorname{KL}\left(\pi, \pi^k\right);$$

$$\pi^{k+1} \leftarrow \underset{\pi}{\arg\max} \mathbb{E}_{s\sim\rho_{\pi^k}}\left[\mathbb{E}_{a\sim\pi}\left[A^{\pi^k}(s,a)\right] - \gamma^k \operatorname{KL}\left(\pi, \pi^k\right)\right],$$

where $\operatorname{KL}(\cdot, \cdot)$ denotes the Bregman divergence corresponding to negative entropy.

# B RELATED WORKS

## B.1 TACKLE NON-STATIONARITY IN MARL

Many works have been proposed to tackle non-stationarity in MARL. These methods range from using a modification of standard RL training schemes to computing and sharing additional other agents' information.

### B.1.1 MODIFICATION OF STANDARD RL TRAINING SCHEMES

One modification of the standard RL training schemes is the centralized critic and decentralized actor (CCDA) architecture. Since the centralized critics can access the information of all other agents during training, the dynamics of the environment remain stable for the agent. Lowe et al. (2017) combined DDPG (Lillicrap et al., 2016) with CCDA architecture and proposed MADDPG algorithm. Foerster et al. (2018b) proposed a counterfactual baseline in the advantage estimation and used the REINFORCE (Williams, 1992) algorithm as the backbone in the CCDA architecture. Similar with Lowe et al. (2017), Iqbal & Sha (2019) combined SAC (Haarnoja et al., 2018) with CCDA architecture and introduced attention mechanism into the centralized critic design. In addition, Iqbal & Sha (2019) also introduced the counterfactual baseline proposed by Foerster et al. (2018b). CCDA alleviated non-stationarity problems indirectly makes it unstable and ineffective, and our experiments have also verified this. Specifically, on the one hand, the centralized critic does not directly affect the agent's policy but only influences the update direction of the policy through the gradient. The policy modeling does not take into account the actions of other agents like centralized critics (considering the decisions of other agents in centralized critics is the crucial improvement of this type of algorithm) but only based on local observations. On the other hand, and more importantly, even if centralized critics explicitly consider the actions of other agents, they cannot mitigate the adverse effects of non-stationarity. Centralized critics only consider the sample of other agent's policy distribution. Once the other agent's policy change drastically and frequently (corresponding to more severe non-stationarity), they need more samples to "implicit" modeling the outside environment, which leads to higher sample complexity.

Another modification to handle non-stationarity in MARL is self-play for competitive tasks or population-based training for cooperative tasks (Papoudakis et al., 2019). Tesauro (1995) used self-play to train the TD-Gammon, which managed to win against the human champion in Backgammon. Baker et al. (2019) extended self-play to more complex environments with continuous state and action space. Liu et al. (2019) and Jaderberg et al. (2019) combined population-based training with self-play to solve complex team competition tasks, a popular 3D multiplayer first-person video game, *Quake III Arena Capture the Flag*, and *MuJoCo Soccer*, respectively. However, such methods require a lot of hardware resources and a well-designed parallelization platform.

### B.1.2 COMPUTING AND SHARING ADDITIONAL INFORMATION

In addition to modifying the standard RL training scheme, there are also methods to solve non-stationarity problems by computing and sharing additional information. One naive approach is parameter sharing and use agents' aggregated trajectories to conduct policy optimization at every iteration (Gupta et al., 2017; Terry et al., 2020). Unfortunately, this simple approach has significant drawbacks. An obvious demerit is that parameter sharing requires that all agents have identical action spaces, *i.e.*, $\mathcal{A}^i = \mathcal{A}^j, \forall i, j \in \mathcal{N}$, which limits the class of MARL problems to solve. Importantly, enforcing parameter sharing is equivalent to putting a constraint $\theta^i = \theta^j, \forall i, j \in \mathcal{N}$ on the joint policy space. In principle, this can lead to a suboptimal solution. To elaborate, we have following proposition proposed by Kuba et al. (2021):

**Proposition 1** (suboptimal)**.** *Let's consider a fully-cooperative game with an even number of agents $n$, one state, and the joint action space $\{0, 1\}^n$, where the reward is given by $r(\mathbf{0}^{n/2}, \mathbf{1}^{n/2}) = r(\mathbf{1}^{n/2}, \mathbf{0}^{n/2}) = 1$, and $r(a^{1:n}) = 0$ for all other joint actions. Let $\mathcal{J}^*$ be the optimal joint reward, and $\mathcal{J}^*_{share}$ be the optimal joint reward under the shared policy constraint. Then*

$$\frac{\mathcal{J}^*_{share}}{\mathcal{J}^*} = \frac{2}{2^n}.$$

This proposition shows that parameter sharing can lead to a suboptimal outcome that is exponentially-worse with the increasing number of agents.

In addition, there are many other ways to share or compute additional information among agents. Foerster et al. (2017) proposed importance sampling corrections to adjust the weight of previous experience to the current environment dynamic to stabilize multi-agent experience replay. Raileanu et al. (2018) and Rabinowitz et al. (2018) used additional networks to predict the actions or goals of other agents and input them as additional information into the policy network to assist decision-making. Foerster et al. (2018a) accessed the optimized trajectory of other agents by explicitly predicting the parameter update of other agents when calculating the policy gradient, thereby alleviating the non-stationarity problem. These explicitly considering other agents' information are also called *modeling of others*. To solve the convergence problem shown by Foerster et al. (2018a) under certain n-player and non-convex games, Letcher et al. (2019) presents Stable Opponent Shaping(SOS). This new method interpolates between Foerster et al. (2018a) and a stable variant named LookAhead, which is proved that converges locally to equilibria and avoids strict saddles in all differentiable games. Different from Foerster et al. (2018a); Letcher et al. (2019) directly predicting the opponent's policy parameters, Xie et al. (2020) solves the non-stationary problem by predicting and influencing the latent representation of the opponent's policy. Recently, Al-Shedivat et al. (2018) transformed non-stationarity problems into meta-learning problems, and extended MAML (Finn et al., 2017) to MAS to find an initialization policy that can quickly adapt to non-stationarity. However, Al-Shedivat et al. (2018) treats other agents as if they are external factors whose learning it cannot affect, and this does not hold in practice for the general multi-agent learning settings. Kim et al. (2021) combines Foerster et al. (2018a) to consider both an agent's own non-stationary policy dynamics and the non-stationary policy dynamics of other agents in the environment. Due to the unique training mechanism of the above methods, they are difficult to extend to the tasks of more than 2 agents.

## B.2    TRUST-REGION METHODS

### B.2.1    TRUST-REGION METHODS IN SINGLE-AGENT RL

*trust-region* or *proximity-based* methods, resonating the fact they make the new policy lie within a trust-region around the old one. Such methods include traditional dynamic programming-based conservative policy iteration (CPI) algorithm (Kakade & Langford, 2002), as well as deep RL methods, such as trust-region policy optimization (TRPO) (Schulman et al., 2015) and proximal policy optimization (PPO) (Schulman et al., 2017). TRPO used line-search to ensure that the KL divergence between the new policy and the old policy is below a certain threshold. PPO is to solve a more relaxed unconstrained optimization problem, in which the ratio of the old and new policy is clipped to a specific bound. Wu et al. (2017) (ACKTR) extended the framework of natural policy gradient and proposed to optimize both the actor and the critic using Kronecker-factored approximate curvature (K-FAC) with trust-region. Nachum et al. (2018) proposed an off-policy trust-region method, Trust-PCL, which introduced relative entropy regularization to maintain optimization stability while exploiting off-policy data. Recently, Tomar et al. (2020) used mirror decent to solve a relaxed unconstrained optimization problem and achieved strong performance.

### B.2.2    TRUST-REGION METHODS IN MARL

Extending trust-region methods to MARL is highly non-trivial. Despite empirical successes, none of them managed to propose a theoretically-justified trust-region protocol in multi-agent learning. Instead, they tend to impose certain assumptions to enable direct implementations of TRPO/PPO in MARL problems. For example, IPPO (de Witt et al., 2020) assume homogeneity of action spaces for all agents and enforce parameter sharing which is discussed above. Yu et al. (2021) proposed MAPPO which enhances IPPO by considering a joint critic function and finer implementation techniques for on-policy methods. Yet, it still suffers similar drawbacks of IPPO. Wen et al. (2021) adjusted PPO for MARL by considering a game-theoretical approach at the meta-game level among agents. Unfortunately, it can only deal with two-agent cases due to the intractability of Nash equilibrium. Hu & Hu (2021) developed Noisy-MAPPO that targets to address the sub-optimality issue; however, it still lacks theoretical insights for the modification made on MAPPO. Recently, Li & He (2020) tried to implement TRPO for MARL through distributed consensus optimization; however, they enforced the same trust-region for all agents (see their Equation (7)) which, similar to parameter sharing, largely limits the policy space for optimization, and this also will make the algorithm face the

trust-region decomposition dilemma. The method HATRPO (Kuba et al., 2021) based on sequential update scheme is proposed from the perspective of monotonic improvement guarantee. But at the same time, sequential update scheme also limit the scalability of the algorithm.

In addition, Jiang & Lu (2021) proposes a MARL algorithm to adjust the learning rates of agents adaptively. Limiting the size of the trust-region of each agent's local policy is related to limiting the learning rate of its local policy. They focus on speeding up the learning speed, instead of solving non-stationarity problems, However, Jiang & Lu (2021) has no theoretical support, and matching the direction of adjusting the learning rates with the directions of maximizing the Q values may cause overestimation problems.

## C    Pseudo-code of MAMT and MAMD

This section gives the pseudo-code of the MAMT algorithm (see Algorithm 2[‖]) and the MAMT algorithm without trust-region decomposition network (see Algorithm 3). For convenience, we named the latter MAMD. In the MAMD algorithm, the trust-region constraint is equally distributed to the local policies of all agents.

---

**Algorithm 2** MAMT

---

**Require:** Initialize behavior policy $\pi_{\psi_i}$ and main centralized critic $Q_{\zeta_i}$ for each agent $i$, empty replay buffer $\mathcal{D}$, trust-region decomposition network $f_{\theta-}$ and $g_{w+}$, local trust-region $\delta_i^0$
1: Set target critic $Q_{\bar{\zeta}_i}$ equal to main critic
2: Set $\pi_{\bar{\psi}_i}$ and $\pi_{\psi_i^{\mathrm{old}}}$ equal to behavior policy
3: **while** not convergence **do**
4:     Observe local observation $o_i$ and select action $a_i \sim \pi_\delta(\cdot \mid o_i)$ for each agent $i$
5:     Execute joint action $a$ in the environment
6:     Observe next local observation $o_i'$, local reward $r_i$ and local done signal $d_i$ of each agent $i$
7:     Store $(\{o_i\}, \{a_i\}, \{r_i\}, \{o_i'\}, \{d_i\})$ in replay buffer $\mathcal{D}$
8:     **if** it's time to update **then**
9:         **for** j in range(however many updates) **do**
10:            Sample a batch of transitions $\mathcal{B}$ from $\mathcal{D}$
11:            Compute $\{\mathcal{C}_{i,\backslash i}^t\}$ with Eq. 5
12:            Compute $\{\mathcal{D}_{i,\mathrm{ns}}^t\}$ with Eq. 6
13:            **for** slow update **do**
14:                Update $\delta_i^t$ with Eq. 8
15:            **for** fast update **do**
16:                Update $f_{\theta-}, g_{w+}$ with Eq. 8
17:            Update centralized critic of all agents by Eq. 9
18:            Update individual policy of all agents by Eq. 10
19:            Update target networks $Q_{\bar{\zeta}_i}$ and $\pi_{\bar{\psi}_i}$
20:            **if** it's time to update **then**
21:                Update old policy $\pi_{\psi_i^{\mathrm{old}}}$

---

**Algorithm 3** MAMD

---

**Require:** Initialize behavior policy $\pi_{\psi_i}$ and main centralized critic $Q_{\zeta_i}$ for each agent $i$, empty replay buffer $\mathcal{D}$
1: Set target critic $Q_{\bar{\zeta}_i}$ equal to main critic
2: Set target policy $\pi_{\bar{\psi}_i}$ equal to behavior policy
3: Set old policy $\pi_{\psi_i^{\mathrm{old}}}$ equal to behavior policy
4: **while** not convergence **do**
5:     Observe local observation $o_i$ and select action $a_i \sim \pi_\delta(\cdot \mid o_i)$ for each agent $i$
6:     Execute joint action $a$ in the environment
7:     Observe next local observation $o_i'$, local reward $r_i$ and local done signal $d_i$ of each agent $i$
8:     Store $(\{o_i\}, \{a_i\}, \{r_i\}, \{o_i'\}, \{d_i\})$ in replay buffer $\mathcal{D}$
9:     **if** All $d_i$ are terminal **then**
10:        Reset the environment
11:     **if** it's time to update **then**
12:         **for** j in range(however many updates) **do**
13:            Sample a batch of transitions $\mathcal{B}$ from $\mathcal{D}$
14:            Update centralized critic of all agents by Eq. 9
15:            Update individual policy of all agents by Eq. 10
16:            Update target networks $Q_{\bar{\zeta}_i}$ and $\pi_{\bar{\psi}_i}$
17:            **if** it's time to update **then**
18:                Update old policy $\pi_{\psi_i^{\mathrm{old}}}$

---

[‖]The source code is available at `https://anonymous.4open.science/r/MAMT`.

## D   TRUST-REGION DECOMPOSITION DILEMMA

In order to verify the existence of the trust-region decomposition dilemma, we defined a simple coordination environment. We extend the *Spread* environment of Section 4 to 3 agents and 3 landmarks, labeled **Spread-3**. We define different Markov random fields (see Figure 7) of the joint policy by changing the reward function to *influence the transition function of each agent indirectly*.

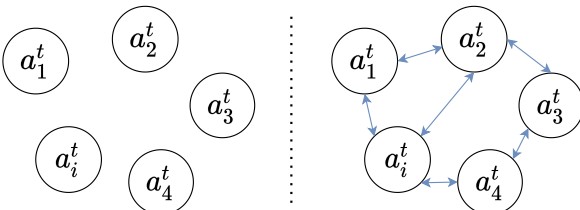

Figure 6: The probabilistic graphical models of two different modelings of the joint policy. **Left:** Modeling the joint policy with the mean-field variation family; **Right:** Modeling the joint policy as a pairwise Markov random field. Each node represents the action of agent $i$ at timestep $t$.

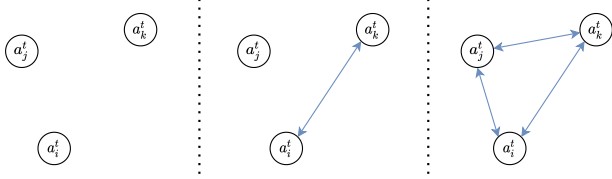

Figure 7: In the Spread-3 environment, 3 different Markov random fields are generated due to the different definitions of the reward function of each agent. Note that these are not all possible Markov random fields, but three typical cases.

Specifically, the reward function of each agent is composed of two parts: the minimum distance between all agents and the landmarks, and the other is the collision. For the leftmost MRF in Figure 7, all agents are independent. The reward function of each agent is only related to itself, only related to the minimum distance between itself and a specific landmark, and will not collide with other agents. We labeled this situation as **Spread-3-Sep**. For the middle MRF in Figure 7, the reward function of agent $j$ is the same as that of *Spread-3-Sep*, which is only related to itself; but agent $i$ and $k$ are interdependent. For agent $i$, its reward function consists of the minimum distance between $i$ and $k$ and the landmark and whether the two collide. The reward function of agent $k$ is similar. We labeled this situation as **Spread-3-Mix**. Finally, the rightmost MRF is consistent with the standard environment settings, labeled **Spread-3-Ful**.

To verify the existence of the trust-region decomposition dilemma, we compare the performance of three different algorithms. First, we select MAAC without any trust-region constraints as the baseline, labeled **MAAC**. Secondly, we choose MAMD based on mean-field approximation and naive trust-region decomposition as one of the algorithms to be compared, labeled **MAMD**. Finally, we optimally assign trust-region based on prior knowledge. For *Spread-3-Sep*, we do not impose any trust-region constraints, same as *MAAC*; for *Spread-3-Mix*, we only impose equal size constraints on agent $i$ and $k$; and for *Spread-3-Ful*, we impose equal size constraints on all agents, same as *MAMD*. We labeled these optimally decomposition as **MAMD-OP**. The performance is shown in Figure 8.

It can be seen from the figure that inappropriate decomposition of the trust-region will negatively affect the convergence speed and performance of the algorithm. The optimal decomposition method can make the algorithm performance and convergence speed steadily exceed baselines.

Note that the three algorithms compared here are all based on the MAAC with centralized critics. After we change the reward function of the agent, the information received by these centralized critics has more redundancy in some scenarios (for example, in *Spread-3-Sep* and *Spread-3-Mix*). To exclude the algorithm's performance from being affected by this redundant information, we output the attention weights in MAAC, as shown in Figure 9. It can be seen from the figure that

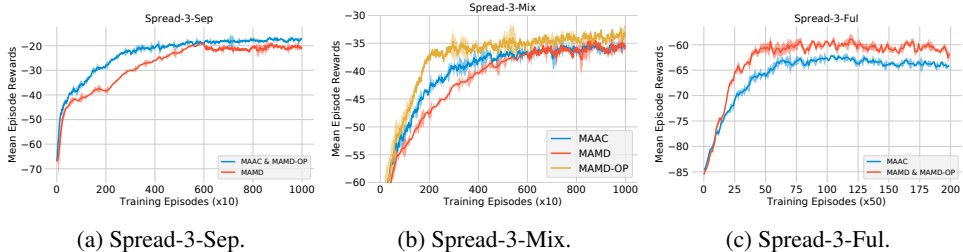

(a) Spread-3-Sep.    (b) Spread-3-Mix.    (c) Spread-3-Ful.

Figure 8: The performance of different trust-region decompositions in different scenarios. These results indicate the existence of a trust-region decomposition dilemma.

different algorithms can filter redundant information well, thus eliminating the influence of redundant information on the convergence speed and performance of the algorithm.

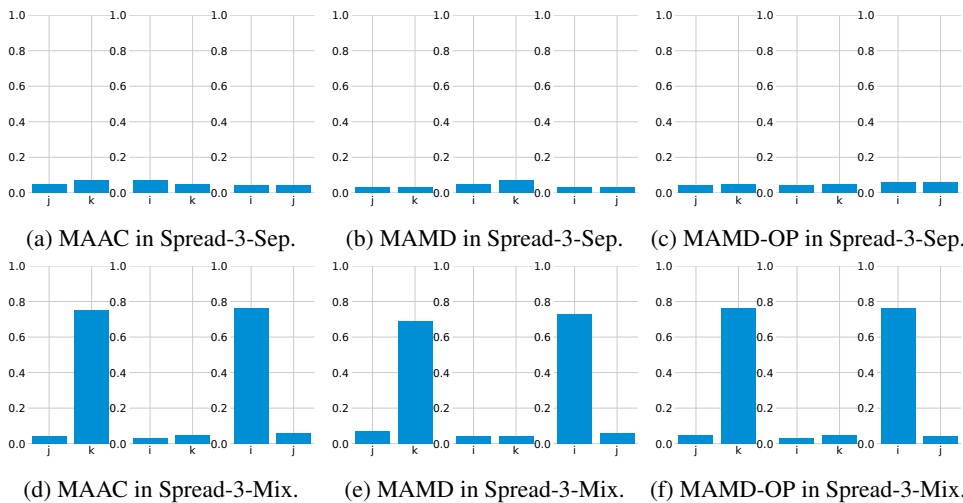

(a) MAAC in Spread-3-Sep. (b) MAMD in Spread-3-Sep. (c) MAMD-OP in Spread-3-Sep.

(d) MAAC in Spread-3-Mix. (e) MAMD in Spread-3-Mix. (f) MAMD-OP in Spread-3-Mix.

Figure 9: The attention weights of the different agents of different trust-region decomposition in different scenarios. These results indicate the existence of a trust-region decomposition dilemma.

# E PROOFS

## E.1 PROOF OF LEMMA 1

*Proof.*

$$
\begin{aligned}
& D_{\mathrm{TV}}(p^t(\cdot|\boldsymbol{o}, a_i), p^{t+1}(\cdot|\boldsymbol{o}, a_i)) \\
&= \max_{\boldsymbol{o}, a_i} |p^{t+1}(\cdot|\boldsymbol{o}, a_i) - p^t(\cdot|\boldsymbol{o}, a_i)| \\
&= \max_{\boldsymbol{o}, a_i} \left| \int p(\cdot|\boldsymbol{o}, \boldsymbol{a}) \cdot \left( \pi_{-i}^{t+1}(a_{-i}|\boldsymbol{o}) - \pi_{-i}^t(a_{-i}|\boldsymbol{o}) \right) \mathrm{d}a_{-i} \right| \\
&\leq \max_{\boldsymbol{o}} \int \left| \pi_{-i}^{t+1}(a_{-i}|\boldsymbol{o}) - \pi_{-i}^t(a_{-i}|\boldsymbol{o}) \right| \mathrm{d}a_{-i} = \max_{\boldsymbol{o}} \left\| \pi_{-i}^{t+1}(\boldsymbol{o}) - \pi_{-i}^t(\boldsymbol{o}) \right\|_1 \\
&\leq 2\ln 2 \cdot \max_{\boldsymbol{o}} D_{\mathrm{KL}}^{1/2}(\pi_{-i}^t(\boldsymbol{o}) \| \pi_{-i}^{t+1}(\boldsymbol{o})) \leq 2\ln 2 \cdot \delta_i^{1/2}.
\end{aligned}
$$

$\square$

## E.2 PROOF OF LEMMA 2

*Proof.*

$$
\begin{aligned}
D_{\mathrm{TV}}(r^t(\boldsymbol{o}, a_i), r^{t+1}(\boldsymbol{o}, a_i)) &= \max_{\boldsymbol{o}, a_i} |r^{t+1}(\boldsymbol{o}, a_i) - r^t(\boldsymbol{o}, a_i)| \\
&= \max_{\boldsymbol{o}, a_i} \left| \int r(\boldsymbol{o}, \boldsymbol{a}) \cdot \left( \pi_{-i}^{t+1}(a_{-i}|\boldsymbol{o}) - \pi_{-i}^t(a_{-i}|\boldsymbol{o}) \right) \mathrm{d}a_{-i} \right| \\
&\leq \max_{\boldsymbol{o}} \int \left| \pi_{-i}^{t+1}(a_{-i}|\boldsymbol{o}) - \pi_{-i}^t(a_{-i}|\boldsymbol{o}) \right| \mathrm{d}a_{-i} \\
&= \max_{\boldsymbol{o}} \left\| \pi_{-i}^{t+1}(\boldsymbol{o}) - \pi_{-i}^t(\boldsymbol{o}) \right\|_1 \\
&\leq 2\ln 2 \cdot \max_{\boldsymbol{o}} D_{\mathrm{KL}}^{1/2}(\pi_{-i}^t(\boldsymbol{o}) \| \pi_{-i}^{t+1}(\boldsymbol{o})) \\
&\leq 2\ln 2 \cdot \delta_i^{1/2}.
\end{aligned}
$$

$\square$

## E.3 PROOF OF THEOREM 1

*Proof.* In this paper, we model the learning procedure of each agent in a multi-agent system as a dynamic non-stationary MDP. From each agent's perspective, the quantities $r_t(\boldsymbol{o}, a_i)$'s and $p_t(\cdot|\boldsymbol{o}, a_i)$'s of each agent $i$ vary across different $t$'s in general. Following Besbes et al. (2014), Cheung et al. (2019) and Mao et al. (2021), we quantify the variations on $r_t(\boldsymbol{o}, a_i)$'s and $p_t(\cdot|\boldsymbol{o}, a_i)$'s in terms of their respective *variation budgets* $B_r, B_p$ $(> 0)$:

$$
B_r = \sum_{t=1}^{T-1} B_{r,t}, \text{ where } B_{r,t} = \max_{o \in \mathcal{O}, a_i \in \mathcal{A}_i} |r_{t+1}(\boldsymbol{o}, a_i) - r_t(\boldsymbol{o}, a_i)|,
$$

$$
B_p = \sum_{t=1}^{T-1} B_{p,t}, \text{ where } B_{p,t} = \max_{o \in \mathcal{O}, a_i \in \mathcal{A}_i} \|p_{t+1}(\cdot \mid \boldsymbol{o}, a_i) - p_t(\cdot \mid \boldsymbol{o}, a_i)\|_1.
$$

To measure the convergence to the best-response from each agent's perspective, we consider an objective of minimizing the *dynamic regret* (Jaksch et al., 2010; Besbes et al., 2014; Cheung et al., 2019; Mao et al., 2021)

$$
\text{Dyn-Reg}_T(\Pi) = \sum_{t=1}^{T} \{\rho_t^* - \mathbb{E}[r_{i,t}(\boldsymbol{o}_t, \boldsymbol{a}_t)]\}.
$$

In the oracle $\sum_{t=1}^{T} \rho_t^*$, the summand $\rho_t^*$ is the optimal long-term average reward of the stationary MDP, i.e., other agents follow the fixed optimal policies, with state transition distribution $p_{i,t}$ and mean reward $r_{i,t}$. Below we give a definition and an assumption.

**Definition 4** (Communicating MDPs and Diameter). *Consider a set of states $\mathcal{S}$, a collection $\mathcal{A} = \{\mathcal{A}_s\}_{s \in \mathcal{S}}$ of action sets, and a state transition distribution $\bar{p} = \{\bar{p}(\cdot \mid s, a)\}_{s \in \mathcal{S}, a \in \mathcal{A}_s}$. For any $s, s' \in \mathcal{S}$ and stationary policy $\pi$, the hitting time from $s$ to $s'$ under $\pi$ is the random variable $\Lambda(s' \mid \pi, s) := \min\{t : s_{t+1} = s', s_1 = s, s_{\tau+1} \sim \bar{p}(\cdot \mid s_\tau, \pi(s_\tau)) \forall \tau\}$, which can be infinite. We say that is a communicating MDP iff $D := \max_{s, s' \in \mathcal{S}} \min_{\text{stationary } \pi} \mathbb{E}[\Lambda(s' \mid \pi, s)]$ is finite. The quantity $D$ is the diameter (Jaksch et al., 2010) associated with $(\mathcal{S}, \mathcal{A}, \bar{p})$.*

**Assumption 2** (Bounded Diameters). *For each $t \in [T]$, the tuple $(\mathcal{S}, \mathcal{A}, p_t)$ constitutes a communicating MDP with diameter at most $D_t$. We denote the maximum diameter as $D_{max} = \max_{t \in \{1, \ldots, T\}} D_t$.*

Then we have following proposition (Cheung et al., 2019) from each agent's perspective:

**Proposition 2.** *Consider an instance $(\mathcal{S}, \mathcal{A}, T, p, r)$ from each agent's perspective that satisfies Assumption 2 with maximum diameter $D_{max}$ and has variation budgets $B_r$, $B_p$ for rewards and transition distributions respectively. In addition, suppose that $T \geq B_r + 2D_{\max} B_p > 0$, then it holds that*

$$\sum_{t=1}^{T} \rho_t^* \geq \max_{\Pi} \left\{ \mathbb{E}\left[ \sum_{t=1}^{T} r_t\left(s_t^{\Pi}, a_t^{\Pi}\right) \right] \right\} - 4(D_{\max} + 1)\sqrt{(B_r + 2D_{\max} B_p) T}.$$

*The maximum is taken over all non-anticipatory policies $\Pi$'s. We denote $\left\{\left(s_t^{\Pi}, a_t^{\Pi}\right)\right\}_{t=1}^{T}$ as the trajectory under policy $\Pi$, where $a_t^{\Pi} \in \mathcal{A}_{s_t^{\Pi}}$ is determined based on $\Pi$ and $\mathcal{H}_{t-1} \cup \{s_t^{\Pi}\}$, and $s_{t+1}^{\Pi} \sim p_t\left(\cdot \mid s_t^{\Pi}, a_t^{\Pi}\right)$ for each $t$.*

The proof of Proposition 2 is shown in (Cheung et al., 2019). Based on Lemma 1 and Lemaa 2, we can easily obtain $B_r \leq 2 \ln 2 \cdot \delta_i^{1/2} \cdot T$ and $B_p \leq 2 \ln 2 \cdot \delta_i^{1/2} \cdot T \cdot |\mathcal{O}|$. Then we have following corollary:

**Corollary 1.** *Consider an instance $(\mathcal{S}, \mathcal{A}, T, p, r)$ from each agent's perspective that satisfies Assumption 2 with maximum diameter $D_{max}$ and has variation budgets $B_r$, $B_p$ for rewards and transition distributions respectively. In addition, suppose that $T \geq B_r + 2D_{\max} B_p > 0$, then it holds that*

$$\sum_{t=1}^{T} \rho_t^* \geq \max_{\Pi} \left\{ \mathbb{E}\left[ \sum_{t=1}^{T} r_t\left(s_t^{\Pi}, a_t^{\Pi}\right) \right] \right\} - 4(D_{\max} + 1) \cdot T\sqrt{(1 + 2 \cdot D_{max}|\mathcal{O}|)2\ln 2 \cdot \delta_i^{1/2}}.$$

According to the Definition 4, Theorem 1 is proved.

$\square$

## E.4 Proof of Theorem 2

*Proof.*

$$
\begin{aligned}
\text{KL}[\pi \mid \pi'] &= \int \int \pi(a_i, a_{-i}|\boldsymbol{o}) \log \frac{\pi(a_i, a_{-i}|\boldsymbol{o})}{\pi'(a_i, a_{-i}|\boldsymbol{o})} \mathrm{d}a_i \mathrm{d}a_{-i} \\
&= \int \int \pi(a_i|a_{-i}, \boldsymbol{o})\pi(a_{-i}|\boldsymbol{o}) \log \frac{\pi(a_i|a_{-i}, \boldsymbol{o})\pi(a_{-i}|\boldsymbol{o})}{\pi'(a_i|a_{-i}, \boldsymbol{o})\pi(a_{-i}|s)} \mathrm{d}a_i \mathrm{d}a_{-i} \\
&= \int \int \pi(a_i|a_{-i}, \boldsymbol{o})\pi(a_{-i}|\boldsymbol{o}) \log \frac{\pi(a_i|a_{-i}, \boldsymbol{o})}{\pi'(a_i|a_{-i}, \boldsymbol{o})} \mathrm{d}a_i \mathrm{d}a_{-i} \\
&\quad + \int \int \pi(a_i|a_{-i}, \boldsymbol{o})\pi(a_{-i}|\boldsymbol{o}) \log \frac{\pi(a_{-i}|\boldsymbol{o})}{\pi'(a_{-i}|\boldsymbol{o})} \mathrm{d}a_i \mathrm{d}a_{-i} \\
&= \int \int \pi(a_i|a_{-i}, \boldsymbol{o})\pi(a_{-i}|\boldsymbol{o}) \log \frac{\pi(a_i|a_{-i}, \boldsymbol{o})}{\pi'(a_i|a_{-i}, \boldsymbol{o})} \mathrm{d}a_i \mathrm{d}a_{-i} + \text{KL}\left[\pi_{-i} \mid \pi'_{-i}\right] \\
&= \int \text{KL}\left[\pi_i(a_{-i}, \boldsymbol{o}) \mid \pi'_i(a_{-i}, \boldsymbol{o})\right] \pi(a_{-i}|\boldsymbol{o}) \mathrm{d}a_{-i} + \text{KL}\left[\pi_{-i} \mid \pi'_{-i}\right] \\
&\geq \text{KL}\left[\pi_{-i} \mid \pi'_{-i}\right].
\end{aligned}
$$

So we have $\mathrm{KL}\left[\pi \mid \pi'\right] \geq \frac{1}{n} \sum_i \mathrm{KL}\left[\pi_{-i} \mid \pi'_{-i}\right]$. Take the maximum value on both sides of the inequality, Theorem 2 is proved.

$\square$

Since a similar conclusion is reached in Li & He (2020), we will make a brief comparison with it here. Theorem 2 establishes the relationship between the *maximum* KL-divergence of the consecutive joint policies of all agents and the *maximum* KL-divergence of the consecutive joint policies of other agents. It establishes the theoretical connection between the KL-divergence of the consecutive joint policies of all agents and environmental non-stationarity. The Equation (8) of Li & He (2020) extends the KL divergence constraint of the TRPO algorithm to the multi-agent scenario and establishes the connection between the divergence of the joint policy of all agents and the divergence of local policy of each agent.

### E.5 PROOF OF THEOREM 3

*Proof.*

$$\mathbb{E}_{\boldsymbol{o}\sim\mu}\left[\mathrm{KL}(\boldsymbol{\pi}(\cdot|\boldsymbol{o}), \boldsymbol{\pi}^k(\cdot|\boldsymbol{o}))\right] < \delta$$

$$\iff \mathbb{E}_{\boldsymbol{o}\sim\mu}\left[\int_a \boldsymbol{\pi}(\cdot|\boldsymbol{o}) \log \frac{\boldsymbol{\pi}(\cdot|\boldsymbol{o})}{\boldsymbol{\pi}^k(\cdot|\boldsymbol{o}))}\mathrm{d}a\right] < \delta$$

$$\iff \mathbb{E}_{\boldsymbol{o}\sim\mu}\left[\int_a \boldsymbol{\pi}(\cdot|\boldsymbol{o}) \left(\sum_{i=1}^n \log \frac{\boldsymbol{\pi}_i(\cdot|o_i)}{\boldsymbol{\pi}_i^k(\cdot|o_i))}\right)\mathrm{d}a\right] < \delta$$

$$\iff \mathbb{E}_{\boldsymbol{o}\sim\mu}\left[\sum_{i=1}^n \int_a \boldsymbol{\pi}(\cdot|\boldsymbol{o}) \log \frac{\boldsymbol{\pi}_i(\cdot|o_i)}{\boldsymbol{\pi}_i^k(\cdot|o_i))}\mathrm{d}a\right] < \delta$$

$$\iff \sum_{i=1}^n \mathbb{E}_{\boldsymbol{o}\sim\mu}\left[\int_a \boldsymbol{\pi}(\cdot|\boldsymbol{o}) \log \frac{\boldsymbol{\pi}_i(\cdot|o_i)}{\boldsymbol{\pi}_i^k(\cdot|o_i))}\mathrm{d}a\right] < \delta$$

$$\iff \sum_{i=1}^n \left[\int_{\boldsymbol{o}} \mu(\boldsymbol{o}) \int_a \boldsymbol{\pi}(\cdot|\boldsymbol{o}) \log \frac{\boldsymbol{\pi}_i(\cdot|o_i)}{\boldsymbol{\pi}_i^k(\cdot|o_i))}\mathrm{d}a\mathrm{d}\boldsymbol{o}\right] < \delta.$$

We first simplify the integral term of the inner layer

$$\int_a \boldsymbol{\pi}(\cdot|\boldsymbol{o}) \log \frac{\boldsymbol{\pi}_i(\cdot|o_i)}{\boldsymbol{\pi}_i^k(\cdot|o_i))}\mathrm{d}a$$

$$\iff \int_{a_{\backslash i} \times a_i} \boldsymbol{\pi}_{\backslash i}(\cdot|o_{\backslash i})\pi_i(\cdot|o_i) \log \frac{\boldsymbol{\pi}_i(\cdot|o_i)}{\boldsymbol{\pi}_i^k(\cdot|o_i))}\mathrm{d}a_{\backslash i}\mathrm{d}a_i$$

$$\iff \int_{a_i} \pi_i(\cdot|o_i) \log \frac{\boldsymbol{\pi}_i(\cdot|o_i)}{\boldsymbol{\pi}_i^k(\cdot|o_i))} \left[\int_{a_{\backslash i}} \boldsymbol{\pi}_{\backslash i}(\cdot|o_{\backslash i})\mathrm{d}a_{\backslash i}\right]\mathrm{d}a_i$$

$$\iff \int_{a_i} \pi_i(\cdot|o_i) \log \frac{\boldsymbol{\pi}_i(\cdot|o_i)}{\boldsymbol{\pi}_i^k(\cdot|o_i))}\mathrm{d}a_i \iff \mathrm{KL}\left(\pi_i(\cdot|o_i), \pi_i^k(\cdot|o_i)\right).$$

We replace the original formula with the simplified one

$$\sum_{i=1}^n \left[\int_{\boldsymbol{o}} \mu(\boldsymbol{o}) \int_a \boldsymbol{\pi}(\cdot|\boldsymbol{o}) \log \frac{\boldsymbol{\pi}_i(\cdot|o_i)}{\boldsymbol{\pi}_i^k(\cdot|o_i))}\mathrm{d}a\mathrm{d}\boldsymbol{o}\right] < \delta$$

$$\iff \sum_{i=1}^n \left[\int_{\boldsymbol{o}} \mu(\boldsymbol{o})\mathrm{KL}\left(\pi_i(\cdot|o_i), \pi_i^k(\cdot|o_i)\right)\mathrm{d}\boldsymbol{o}\right] < \delta$$

$$\iff \sum_{i=1}^n \left[\int_{\boldsymbol{o}} \mu(\boldsymbol{o})\delta(o_i)\mathrm{d}\boldsymbol{o}\right] < \delta.$$

where we denote $\mathrm{KL}\left(\pi_i(\cdot|o_i), \pi_i^k(\cdot|o_i)\right)$ as $\delta(o_i)$. For the outer integral term, we have

$$\int_{\boldsymbol{o}} \mu(\boldsymbol{o}) \delta(o_i) \mathrm{d}\boldsymbol{o}$$

$$\iff \int_{o_{\setminus i} \times o_i} \mu(o_{\setminus i}) \mu(o_i) \delta(o_i) \mathrm{d}o_{\setminus i} \mathrm{d}o_i$$

$$\iff \int_{o_i} \mu(o_i) \delta(o_i) \left[\int_{o_{\setminus i}} \mu(o_{\setminus i}) \mathrm{d}o_{\setminus i}\right] \mathrm{d}o_i$$

$$\iff \int_{o_i} \mu(o_i) \delta(o_i) \mathrm{d}o_i$$

$$\iff \mathbb{E}_{o_i \sim u_i}\left[\delta(o_i)\right].$$

Overall, we have

$$\mathbb{E}_{\boldsymbol{o} \sim \mu}\left[\mathrm{KL}(\boldsymbol{\pi}(\cdot|\boldsymbol{o}), \boldsymbol{\pi}^k(\cdot|\boldsymbol{o}))\right] < \delta$$

$$\iff \sum_{i=1}^{n}\left[\int_{\boldsymbol{o}} \mu(\boldsymbol{o}) \int_{a} \boldsymbol{\pi}(\cdot|\boldsymbol{o}) \log \frac{\boldsymbol{\pi}_i(\cdot|o_i)}{\boldsymbol{\pi}_i^k(\cdot|o_i))} \mathrm{d}a \mathrm{d}\boldsymbol{o}\right] < \delta$$

$$\iff \sum_{i=1}^{n}\left[\int_{\boldsymbol{o}} \mu(\boldsymbol{o}) \delta(o_i) \mathrm{d}\boldsymbol{o}\right] < \delta$$

$$\iff \sum_{i=1}^{n} \mathbb{E}_{o_i \sim u_i}\left[\delta(o_i)\right] < \delta$$

$$\iff \sum_{i=1}^{n} \mathbb{E}_{o_i \sim u_i}\left[\mathrm{KL}\left(\pi_i(\cdot|o_i), \pi_i^k(\cdot|o_i)\right)\right] < \delta.$$

$\square$

## F  MORE DISCUSSION ON THE LINEAR REGRET

The bound on Theorem 1 has a linear dependence on the number of time intervals $T = HK$. Therefore, the average regret doesn't vanish to zero as $T$ becomes large, even when small non-stationarity. But from the experimental results, our algorithm has a faster convergence speed and better performance than baselines. Therefore, the rest of this section will expand from the following two aspects. First, although theoretically only linear regret can be achieved, it can be close to sub-linear in actual implementation; second, we can achieve theoretical and implementation sub-linearity through some improvements that will bring a higher computational load.

### F.1  MAMT IS A NEAR-SUBLINEAR REGRET IMPLEMENTATION

| Setting | Algorithm | Regret |
|---|---|---|
| Undis-counted | Jaksch et al. (2010) | $\widetilde{O}(S\ A^{\frac{1}{2}}L^{\frac{1}{3}}D\ T^{\frac{2}{3}})$ |
| | Gajane et al. (2018) | $\widetilde{O}(S^{\frac{2}{3}}A^{\frac{1}{3}}L^{\frac{1}{3}}D^{\frac{2}{3}}T^{\frac{2}{3}})$ |
| | Ortner et al. (2020) | $\widetilde{O}(S\ A^{\frac{1}{2}}\Delta^{\frac{1}{3}}D\ T^{\frac{2}{3}})$ |
| | Cheung et al. (2019) | $\widetilde{O}(S^{\frac{2}{3}}A^{\frac{1}{2}}\Delta^{\frac{1}{4}}D\ T^{\frac{3}{4}})$ |
| Episodic | Domingues et al. (2021) | $\widetilde{O}(S\ A^{\frac{1}{2}}\Delta^{\frac{1}{3}}H^{\frac{4}{3}}T^{\frac{2}{3}})$ |
| | (Mao et al., 2021) | $\widetilde{O}(S^{\frac{1}{3}}A^{\frac{1}{3}}\Delta^{\frac{1}{3}}H\ T^{\frac{2}{3}})$ |
| | (Mao et al., 2021) | $\widetilde{O}(S^{\frac{1}{3}}A^{\frac{1}{3}}\Delta^{\frac{1}{3}}HT^{\frac{2}{3}}+H^{\frac{3}{4}}T^{\frac{3}{4}})$ |

Table 1: Dynamic regret comparisons for single-agent RL in non-stationary MDPs. $S$ and $A$ are the numbers of states and actions, $L$ is the number of abrupt changes, $\Delta = B_r + B_p$, $D$ is the maximum diameter, $H$ is the number of steps per episode, and $T$ is the total number of steps. $\widetilde{O}(\cdot)$ suppresses logarithmic terms.

First, we explain why controlling the $\delta$-stationarity can only achieve linear regret. Table 1 lists the sublinear regret bounds that current SOTA algorithms to solve the single-agent non-stationarity problem can reach. It is worth noting that these single-agent algorithms cannot modify $\Delta$ because this is the intrinsic property of the external environment. But in our method, the environment is made up of other agents' policies, so $\Delta$ can be adjusted. As can be seen from the table, the power of $\Delta = (B_r + B_p)$ and $T$ add up to 1. And from the proof of Theorem 1, $B_r$ and $B_p$ are both the summation over $T$, i.e., $B_r = \sum_{t=1}^{T} B_r^t$ and $B_p = \sum_{t=1}^{T} B_p^t$. However, opponent switching cost considers the maximum value of the continuous policy divergence in the entire learning procedure, so we have

$$\Delta^a T^{1-a} = (B_r + B_p)^a T^{1-a} = (\sum_{t=1}^{T} B_r^t + \sum_{t=1}^{T} B_p^t)^a T^{1-a}$$

$$= (\sum_{t=1}^{T} \delta_i + \sum_{t=1}^{T} \delta_i)^a T^{1-a} = (2\delta_i)^a T,$$

where $0 < a \leq 1/3$. In general, constraining $\delta$-stationarity produces a linear regret because the opponent switching cost is a relatively loose constraint. The key to achieving sub-linear regret is to constrain the consecutive joint policy divergence at each timestep, not the maximum or expected value of the entire learning procedure. Therefore, we propose a tighter constraint based on opponent switching cost, and prove that by imposing this constraint on the consecutive joint policy divergence, the algorithm can achieve sub-linear regret $\tilde{O}(D_{\max}^{3/2}|\mathcal{O}|^{1/2}\Delta_{\delta}^{1/4}T^{1/2})$. First, we define the temporal opponent switching cost.

**Definition 5** (Temporal opponent switching cost). *Let $H$ be the horizon of the MDP and $K$ be the number of episodes that the agent can play, so that total number of steps $T := HK$. The temporal opponent switching cost of agent $i$ is defined as the Kullback–Leibler divergence (over the joint*

Table 2: The proportion of the timesteps that the MAMT meets the condition of a<b to the total timesteps in the training procedure of different cooperative tasks.

| Spread | Multi-Walker | Rover-Tower | Pursuit |
|--------|--------------|-------------|---------|
| 95.47% | 90.16% | 83.76% | 85.88% |

*observation space $\mathcal{O}$ at specific timestep t) between any pair of opponents' joint policies $(\pi_{-i}, \pi'_{-i})$ on which $\pi_{-i}$ and $\pi'_{-i}$ are different:*

$$d^{i,t}_{switch}\left(\pi_{-i}, \pi'_{-i}\right) := \max\left\{\boldsymbol{o} \in \mathcal{O} : D_{\mathrm{KL}}\left(\pi^t_{-i}(\boldsymbol{o}) \| \left[\pi'_{-i}\right]^t(\boldsymbol{o})\right)\right\},$$

*where $\boldsymbol{o}, \mathcal{O}$ are the joint observation and observation space and $t \in T$.*

Correspondingly, we can also define temporal $\delta^t$-stationarity.

**Definition 6** (Temporal $\delta^t$-stationarity). *For a MAS containing $n$ agents, if we have $d^{i,t}_{switch}\left(\pi_{-i}, \pi'_{-i}\right) \leq \delta^t_i$, then the learning procedure of agent $i$ is $\delta^t_i$-stationary at timestep t. Further, if all agents are $\delta$-stationary with corresponding $\{\delta^t_i\}$, then the learning procedure of entire multi-agent system is $\delta^t$-stationary with $\delta^t = \frac{1}{n}\sum_i \delta^t_i$.*

Similarly, we can also get Lemma 1 and Lemma 2 about temporal $\delta^t$-stationarity. Since the content is similar, we won't repeat them here. Then we have $B_r \leq 2\ln 2 \cdot \sum_{t=1}^{T} \delta^{t\,1/2}_i$ and $B_p \leq 2\ln 2 \cdot \sum_{t=1}^{T} \delta^{t\,1/2}_i \cdot |\mathcal{O}|$. In addition, we denote, $B^i_\delta = \sum_{t=1}^{T} \delta^{t\,1/2}_i$. With these definitions and lemmas, we can get the following theorem.

**Theorem 4.** *Let $H$ be the horizon and $K$ be the number of episodes, so that total number of steps $T := HK$. Consider the learning procedure of a MAS satisfies the $\delta^t$-stationarity and each agent satisfies the $\delta^t_i$-stationarity at each timestep $t \in T$. In addition, suppose that $T \geq B_r + 2D_{max}B_p > 0$, then a $\tilde{O}(D^{3/2}_{\max}|\mathcal{O}|^{1/2}B^{i\,1/2}_\delta T^{1/2})$ sublinear dynamic regret bound is attained for each agent $i$.*

The proof is similar to Theorem 1. Theorem 4 shows that if we want to achieve a sub-linear regret, we need to constrain the consecutive joint policy divergence at each timestep instead of only constraining the maximum value of the policy divergence. Looking back at the implementation of the MAMT, although the algorithm has a global hyperparameter $\delta$ to constrain the adaptively adjusted $\{\delta_i\}^N_{i=1}$, the constraint only works when $\sum_i \delta_i > \delta$. We have observed in the experiment that most of the time $\sum_i \delta_i \leq \delta$ (see Table 2). In this way, a different $\delta^t = \sum_i \delta^t_i$ at each timestep $t$ constrain the consecutive joint policy divergence, thereby approximately achieving a sub-linear regret.

### F.2 FUTURE WORK TO ACHIEVE SUBLINEAR REGRET

Again, Theorem 4 shows that if we want to achieve a sub-linear regret, we need to constrain the consecutive joint policy divergence at each timestep. This means that we need to set a hyperparameter $\delta^t$ at every timestep $t$ to constrain the consecutive joint policy divergence, and a manual adjustment will bring an extremely complicated workload.

A more realistic way is to treat the $\delta^t$ of each timestep as an additional optimization variable and formulate it as a dual variable about $delta^t_i$ and the policy parameter $\psi^t_i$. Then the dual gradient descent can be used to optimize $\delta^t$, similar to the adaptive reward scale learning in the soft-actor-critic (SAC, Haarnoja et al. (2018)) algorithm. In addition, non-parameter optimization methods, such as evolutionary methods, can also be used to adjust the $\delta^t$. But no matter which technique is adopted, it will bring a higher computational load. How to balance the computational load and algorithm performance, we leave it to the future to explore.

# G    EXPERIMENTAL DETAILS

## G.1    ENVIRONMENTS

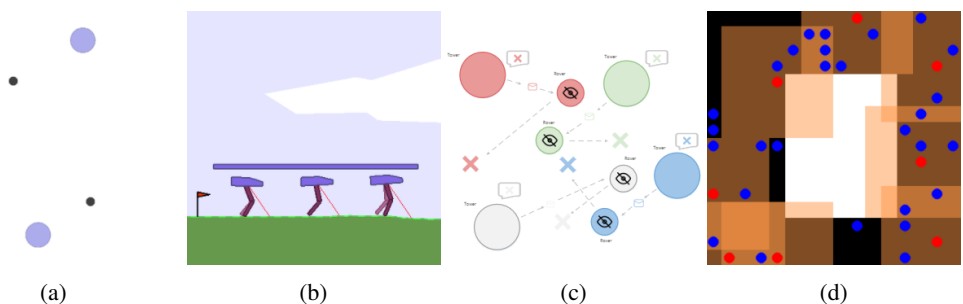

|     |     |     |     |
| :-: | :-: | :-: | :-: |
| (a) | (b) | (c) | (d) |

Figure 10: Coordination environments with increasing complexity.(a) Spread; (b) Multi-Walker; (c) Rover-tower; (d) Pursuit.

**Spread.** This environment (Lowe et al., 2017) has 2 agents, 2 landmarks. Each agent is globally rewarded based on how far the closest agent is to each landmark (sum of the minimum distances). Locally, the agents are penalized if they collide with other agents ($-1$ for each collision).

**Multi-Walker.** In this environment (Terry et al., 2020a), bipedal robots attempt to carry a package as far right as possible. A package is placed on top of 3 bipedal robots. A positive reward is awarded to each walker, which is the change in the package distance.

**Rover-Tower.** This environment (Iqbal & Sha, 2019) involves 8 agents, 4 of which are "rovers" and another 4 which are "towers". In each episode, rovers and towers are randomly paired. The pair is negatively rewarded by the distance of the rover to its goal. The rovers are unable to see in their surroundings and must rely on communication from the towers, sending one of 5 discrete messages.

**Pursuit.** 30 blue evaders and 8 red pursuer agents are placed in a grid with an obstacle. The evaders move randomly, and the pursuers are controlled (Terry et al., 2020a). Every time the pursuers surround an evader, each of the surrounding agents receives a reward of 5, and the evader is removed from the environment. Pursuers also receive a reward of 0.01 every time they touch an evader.

## G.2    OTHER DETAILS

**Random seeds.** All experiments were run for 8 random seeds each. Graphs show the average (solid line) and std dev (shaded) performance over random seed throughout training.

**Performance metric.** Performance for the on-policy (MA-PPO) algorithms is measured as the average reward across the batch collected at each epoch. Performance for the off-policy algorithms (MAMT, MAMD, LOLA, MADDPG, and MAAC) is measured by running the deterministic policy (or, in the case of SAC, the mean policy) without action noise for 10 trajectories and reporting the average reward over those test trajectories.

**Network Architecture.** Figure 11 shows the detailed parameters of the three main networks in the MAMT algorithm.

**Hyperparameters.** Table 3 shows the default configuration used for all the experiments of our methods (MAMD and MAMT) and baselines in this paper. We do not fine-tune the hyperparameters of baselines and use the default setting as same as original papers, see Table 4,5,6 and 7. The hyperparameter fine-tune range of our methods are shown in Table 8 and 9. Considering the long training time of the MARL algorithm, we did not train all hyperparameter combinations to the pre-defined maximum number of episodes for MAMT. We first train all hyperparameter combinations to one-sixth of the maximum number of episodes and select the top-sixth hyperparameter combinations with the best performance (defined in **Performance metric.**). Then we train the selected one-sixth combination to one-third of the maximum number of episodes and select the best-performing one-sixth combination. Finally, we train the remaining combinations to the maximum number of episodes and select the best hyperparameter combination.

Table 3: Default settings of our methods used in experiments.

| Name | Default value |
|---|---|
| num parallel envs | 12 |
| step size | from 10,000 to 50,000 |
| num epochs per step | 4 |
| steps per update | 100 |
| buffer size | 1,000,000 |
| batch size | 1024 |
| batch handling | Shuffle transitions |
| num critic attention heads | 4 |
| value loss | MSE |
| modeling policy loss | CrossEntropyLoss |
| discount | 0.99 |
| optimizer | Adam |
| adam lr | 1e-3 |
| adam mom | 0.9 |
| adam eps | 1e-7 |
| lr decay | 0.0 |
| policy regularization type | L2 |
| policy regularization coefficient | 0.001 |
| modeling policy regularization type | L2 |
| modeling policy regularization coefficient | 0.001 |
| critic regularization type | L2 |
| critic regularization coefficient | 1.0 |
| critic clip grad | 10 * num of agents |
| policy clip grad | 0.5 |
| soft reward scale | 100 |
| modeling policy clip grad | 0.5 |
| trust-region decomposition network clip grad | 10 * num of agents |
| trust-region clip | from 0.01 to 100 |
| num of iteration delay in mirror descent | 100 |
| tsallis q in mirror descent | 0.2 |
| $\delta$ in coordination coefficient | 0.2 |

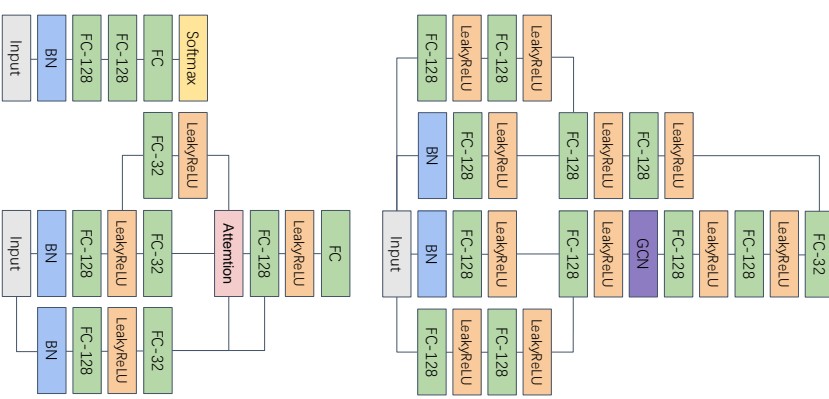

Figure 11: The actor-network (and modeling policy) architecture, critic-network architecture, and trust-region decomposition network architecture from left to right and from top to bottom.

**Hardware.** The hardware used in the experiment is a server with 128G memory and 4 NVIDIA 1080Ti graphics cards with 11G video memory.

**The Code of Baselines.** The code and license of baselines are shown in following list:

Table 4: Default settings of MAAC used in experiments.

| Name | Default value |
|---|---|
| num parallel envs | 12 |
| step size | from 10,000 to 50,000 |
| num epochs per step | 4 |
| steps per update | 100 |
| buffer size | 1,000,000 |
| batch size | 1024 |
| batch handling | Shuffle transitions |
| num critic attention heads | 4 |
| value loss | MSE |
| discount | 0.99 |
| optimizer | Adam |
| adam lr | 1e-3 |
| adam mom | 0.9 |
| adam eps | 1e-7 |
| lr decay | 0.0 |
| policy regularization type | L2 |
| policy regularization coefficient | 0.001 |
| critic regularization type | L2 |
| critic regularization coefficient | 1.0 |
| critic clip grad | 10 * num of agents |
| policy clip grad | 0.5 |
| soft reward scale | 100 |

Table 5: Default settings of MADDPG used in experiments.

| Name | Default value |
|---|---|
| num parallel envs | 12 |
| step size | from 10,000 to 50,000 |
| num epochs per step | 4 |
| steps per update | 100 |
| buffer size | 1,000,000 |
| batch size | 1024 |
| batch handling | Shuffle transitions |
| value loss | MSE |
| discount | 0.99 |
| optimizer | Adam |
| adam lr | 1e-3 |
| adam mom | 0.9 |
| adam eps | 1e-7 |
| lr decay | 0.0 |
| policy regularization type | L2 |
| policy regularization coefficient | 0.001 |
| critic regularization type | L2 |
| critic regularization coefficient | 1.0 |
| critic clip grad | 10 * num of agents |
| policy clip grad | 0.5 |

- MADDPG (Lowe et al., 2017): https://github.com/shariqiqbal2810/maddpg-pytorch, MIT License;

- MAAC (Iqbal & Sha, 2019): https://github.com/shariqiqbal2810/MAAC, MIT License;

- MA-PPO: https://github.com/zoeyuchao/mappo, MIT License;

Table 6: Default settings of MAPPO used in experiments.

| Name | Default value |
|------|---------------|
| num parallel envs | 12 |
| step size | from 10,000 to 50,000 |
| batch size | 4096 |
| num critic attention heads | 4 |
| value loss | Huber Loss |
| Huber delta | 10.0 |
| GAE lambda | 0.95 |
| discount | 0.99 |
| optimizer | Adam |
| adam lr | 1e-3 |
| adam mom | 0.9 |
| adam eps | 1e-7 |
| lr decay | 0.0 |
| policy regularization type | L2 |
| policy regularization coefficient | 0.0 |
| critic regularization type | L2 |
| critic regularization coefficient | 0.0 |
| critic clip grad | 10 |
| policy clip grad | 10 |
| use reward normalization | TRUE |
| use feature normalization | TRUE |

Table 7: Default settings of LOLA(+DQN) used in experiments.

| Name | Default value |
|------|---------------|
| num parallel envs | 12 |
| step size | from 10,000 to 50,000 |
| num epochs per step | 4 |
| steps per update | 100 |
| buffer size | 1,000,000 |
| batch size | 1024 |
| batch handling | Shuffle transitions |
| value loss | MSE |
| discount | 0.99 |
| optimizer | Adam |
| adam lr | 1e-3 |
| adam mom | 0.9 |
| adam eps | 1e-7 |
| lr decay | 0.0 |
| regularization type | L2 |
| regularization coefficient | 1.0 |
| clip grad | 10 * num of agents |

- LOLA (Foerster et al., 2018a): https://github.com/alexis-jacq/LOLA_DiCE and https://github.com/geek-ai/MAgent, MIT License.

Learning curves are smoothed by averaging over a window of 11 epochs. Source code is available at https://anonymous.4open.science/r/MAMT.

Table 8: Tuning ranges of key hyperparameters of MAMD in experiments.

| Name | Range |
|---|---|
| num epochs per step | $\{1, 4, 8\}$ |
| adam lr | $\{0.0003, 0.001\}$ |
| soft reward scale | $\{10, 100\}$ |
| num of iteration delay in mirror descent | $\{100, 1000\}$ |

Table 9: Tuning ranges of key hyperparameters of MAMT in experiments.

| Name | Range |
|---|---|
| num epochs per step | $\{1, 4, 8\}$ |
| adam lr | $\{0.0003, 0.001\}$ |
| soft reward scale | $\{10, 100\}$ |
| trust-region clip | $\{0.01, 1, 100\}$ |
| num of iteration delay in mirror descent | $\{100, 1000\}$ |
| $\delta$ in coordination coefficient | $\{0.002, 0.02, 0.2\}$ |

## H  MORE RESULTS AND ABLATION STUDIES

Figure 13 to Figure 23 show the performance indicators of all agents in 4 environments under different random seeds. In addition, we have also conducted additional ablation studies to analyze the importance of different modules in MAMT. These modules mainly include mirror descent, coordination coefficients, and modeling of others.

**Regarding the mirror descent.**    Considering the low sample efficiency of MARL, we did not adopt the fully on-policy learning but used off-policy training with a small replay buffer. In this way, it can be ensured that samples are less different from the current policy, thereby alleviating the instability caused by off-policy training. The methods such as TRPO and PPO are all on-policy algorithms. Although there are some off-policy trust-region methods, such as Trust-PCL, etc., they are all more complicated. To this end, we adopted a more concise technique, Mirror Descent, as the optimization method of MAMT. MAMT can also use the on-policy method as the backbone, such as PPO. To this end, we have added a comparative experiment, replacing Mirror Descent with the PPO algorithm (MAMT-PPO).

**Regarding the coordination coefficients.**    In the MAMT algorithm, the counterfactual baseline is used in policy learning and used in the calculation of coordination coefficients. To verify the relationship between the correctness of the modeling of the dependencies between agents and the algorithm's performance, we have added two comparative experiments to the revised version. Specifically, we used two pre-defined ways to set the coordination coefficients, set to a fixed value (MAMT-Fixed) of $1/(n-1)$ (where $n$ is the number of agents), and random sampling (MAMT-Random, random sampling from the range of $(0,1)$ and then input into the $\mathrm{softmax}$ function).

**Regarding the modeling of others.**    We borrowed the idea in offline RL to measure the degree of out-of-distribution (see Equation (5)) through the model-based RL to reasonably characterize the non-stationarity or the consecutive joint policy divergence. However, a more intuitive and simple method is to directly perform a weighted summation of the consecutive local policy divergence (the weights are the coordination coefficients). The reason why we did not adopt this scheme is based on the following considerations. It can be seen from Definition 3 that the agents' policies do not directly cause the non-stationarity. The agent cannot directly observe the opponent's policy but only the action sequences. This explicit information directly affects the learning process of the agent and leads to non-stationarity. To verify the correctness of the idea, we also did additional comparative experiments. Specifically, we changed the calculation method of coordination coefficient to the weighted summation of consecutive local policy divergence (MAMT-WS), i.e. $1/n \cdot \sum_i \mathcal{C}_{i,j} \sum_{j \neq i} \mathrm{KL}[\pi'_{\psi_j}(o_j^t) \| \pi_{\psi_j}(o_j^t)]$.

Table 10: The average return of baselines and ablation algorithms in the last $1,000$ episodes of the training stage. The red part represents the newly added experiment results. The numbers in parentheses indicate the standard deviations under 5 (the red part is 3) different random seeds. MAMT-Random stands for randomly setting the coordination coefficients in MAMT, MAMT-Fixed stands for setting the coordination coefficients in MAMT to a fixed value of $1/(n-1)$, MAMT-WS stands for using the weighted sum of the local policy divergence to approximate the joint policy divergence, and MAMT-PPO represents the replacement of the off-policy mirror descent algorithm in MAMT with the on-policy PPO algorithm.

| | Baselines | | | | |
|---|---|---|---|---|---|
| **Environments** | **MAAC** | **MA-PPO** | **MADDPG** | **LOLA** | **MAMT** |
| **Spread** | $-43.23(\pm0.76)$ | $-42.29(\pm0.61)$ | $-43.33(\pm0.68)$ | $-42.08(\pm1.23)$ | $\mathbf{-38.06(\pm0.69)}$ |
| **Multi-Walker** | $-30.16(\pm3.38)$ | $-38.52(\pm4.22)$ | $-49.76(\pm3.97)$ | ✗ | $\mathbf{-4.70(\pm4.38)}$ |
| **Rover-Tower** | $121.79(\pm6.84)$ | $120.36(\pm6.02)$ | $87.46(\pm6.50)$ | ✗ | $\mathbf{145.95(\pm5.77)}$ |
| **Pursuit** | $19.86(\pm1.19)$ | $20.05(\pm1.20)$ | $5.05(\pm1.33)$ | ✗ | $\mathbf{25.62(\pm1.30)}$ |
| | Ablation Studies | | | | |
| **Environments** | **MAMD** | **MAMT-Random** | **MAMT-Fixed** | **MAMT-WS** | **MAMT-PPO** |
| **Spread** | $-42.17(\pm1.05)$ | $-43.04(\pm0.88)$ | $-41.29(\pm0.73)$ | $\mathbf{-39.83(\pm0.72)}$ | $\mathbf{-39.65(\pm0.47)}$ |
| **Multi-Walker** | $\mathbf{-4.92(\pm3.91)}$ | $-30.52(\pm3.17)$ | $\mathbf{-4.82(\pm3.14)}$ | $-4.90(\pm4.86)$ | $-4.98(\pm4.05)$ |
| **Rover-Tower** | $127.80(\pm5.04)$ | $115.11(\pm6.13)$ | $123.21(\pm5.87)$ | $133.70(\pm6.22)$ | $\mathbf{141.33(\pm5.95)}$ |
| **Pursuit** | $22.30(\pm1.81)$ | $16.48(\pm1.33)$ | $21.54(\pm1.62)$ | $17.17(\pm2.89)$ | $\mathbf{25.04(\pm1.68)}$ |

We compared the above ablation algorithms under 4 cooperative tasks, and the experimental results are shown in Table 10. The following points can be seen from the Table 10. First of all, from the

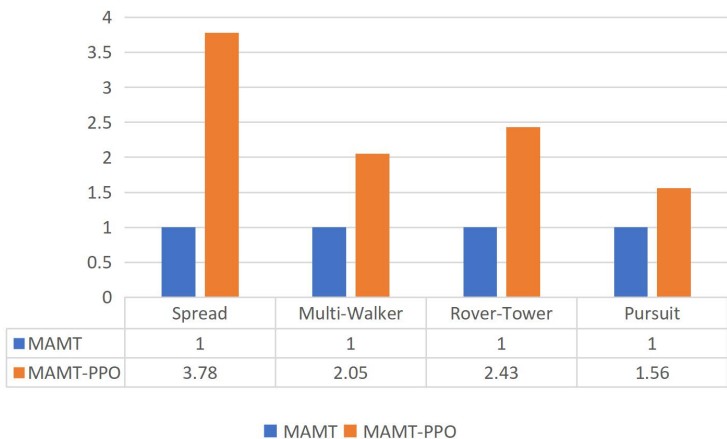

| | Spread | Multi-Walker | Rover-Tower | Pursuit |
|---|---|---|---|---|
| ■ MAMT | 1 | 1 | 1 | 1 |
| ■ MAMT-PPO | 3.78 | 2.05 | 2.43 | 1.56 |

■ MAMT  ■ MAMT-PPO

Figure 12: The proportional relationship between the number of episodes experienced by the MAMT and MAMT-PPO algorithms when they achieve the results in Table 10.

comparison results of MAMT-PPO and MAMT, it can be seen that our method is not significantly affected by the optimization method of the policy learning, and both achieve the best results on the four cooperative tasks. However, MAMT-PPO is an on-policy method, which makes its sample effectiveness lower than MAMT. In Figure 12, we have made statistics on the number of episodes required by the two to achieve the performance in Table 10. It can be seen from the table that MAMT only needs about half of the data to achieve the same performance as MAMT-PPO.

Secondly, from the results of MAMT-Random and MAMT-Fixed, the coordination coefficient is significant to the algorithm's performance, which is also consistent with the theoretical analysis of the trust-region decomposition dilemma in the main body. Here we have observed two interesting phenomena. First, we found that randomizing the coordination coefficient will seriously affect the performance of MAMT. The possible reason is that the wrong trust-region constraint on the local policy will hinder the agent's learning. Second, we found that setting the coordination coefficient to the same fixed value will make the performance of the MAMT algorithm approach that of MAMD. This shows that the TRD-Net fits the relationship between the local trust regions and the approximated consecutive joint policy divergence well.

Third, it can be seen from the results of MAMT-WS that the effect of approximating the KL divergence of consecutive joint policies by using the weighted summation of the KL divergence of consecutive local policy policies is very unstable. In all cooperative tasks, MAMT-WS has greater variance. In simple tasks, such as Spread (2-agents) and Multi-Walker (3-agents), MAMT-WS can have an effect close to MAMT. But once the difficulty of the task increases, the effect of MAMT-WS will even be at the same level as MAMT-Random. We believe that the main reason for this instability is that the agent will cause rapid changes in local policies due to exploration in the early stages of learning.

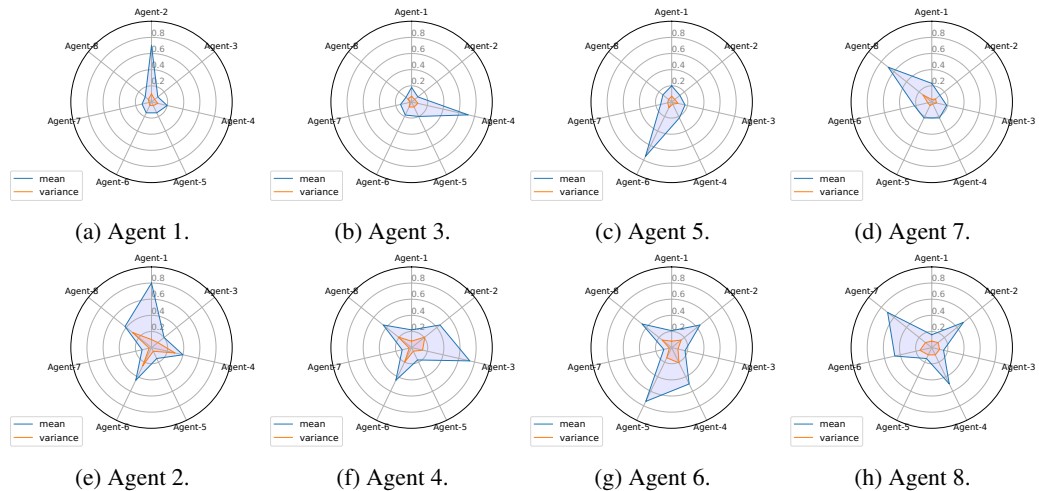

Figure 13: The mean and variance of coordination coefficient of each agent in *Rover-Tower* environment.

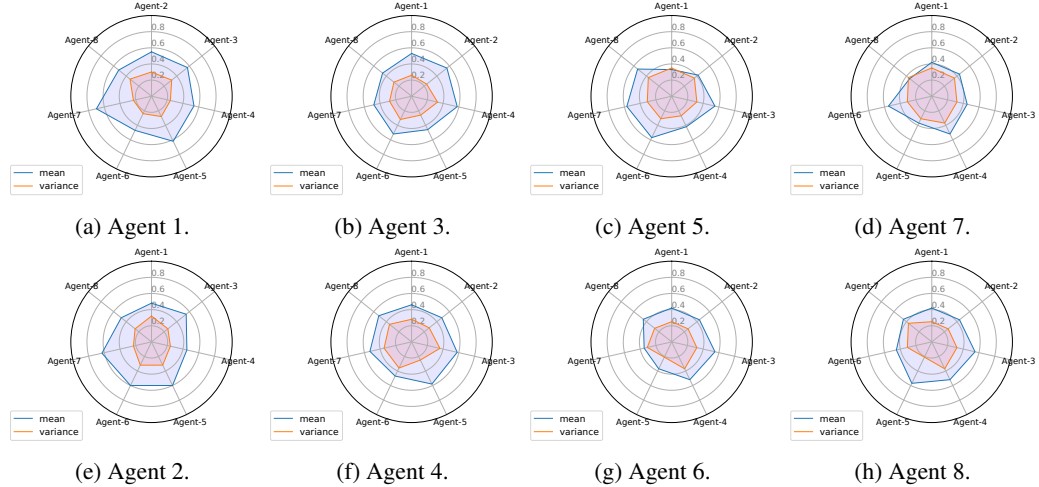

Figure 14: The mean and variance of coordination coefficient of each agent in *Pursuit* environment.

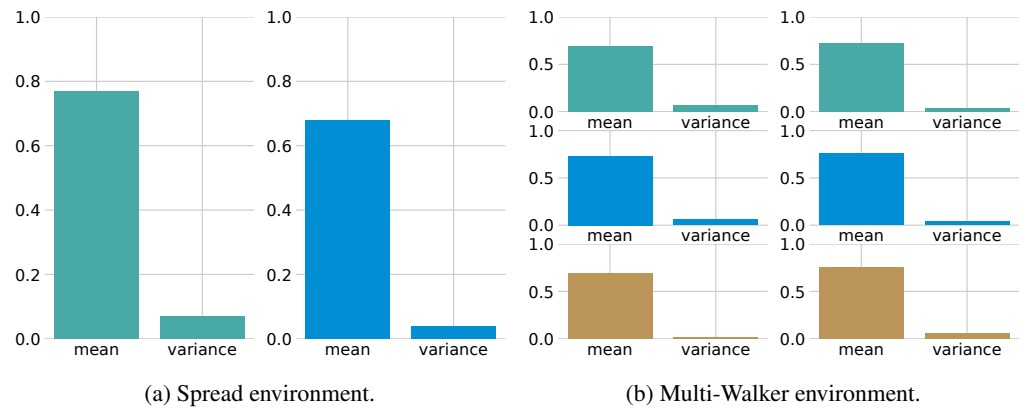

Figure 15: The mean and variance of coordination coefficient of each agent in *Spread* environment and *Multi-Walker* environment. Different color represents different agent.

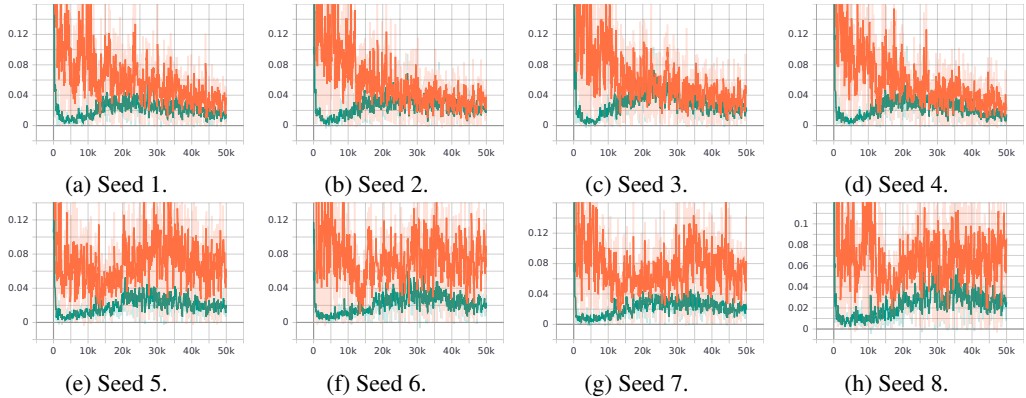

Figure 16: The averaged KL-divergence of each agent in *Rover-Tower* environments. Red line represents MAAC and green line represents MAMT.

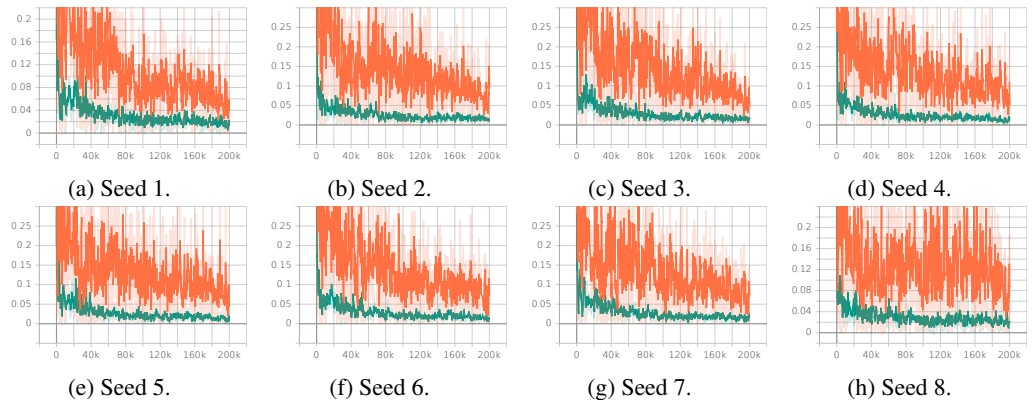

Figure 17: The averaged KL-divergence of each agent in *Pursuit* environments. Red line represents MAAC and green line represents MAMT.

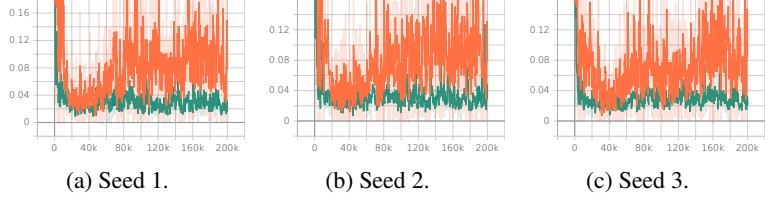

Figure 18: The averaged KL-divergence of each agent in *Multi-Walker* environments. Red line represents MAAC and green line represents MAMT.

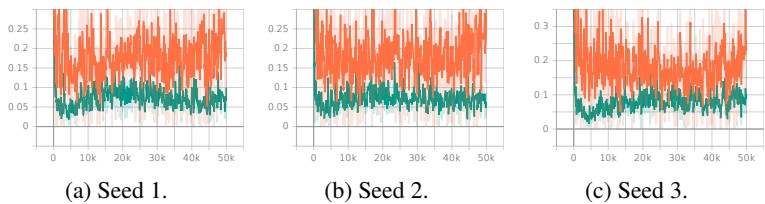

Figure 19: The averaged KL-divergence of each agent in *Spread* environments. Red line represents MAAC and green line represents MAMT.

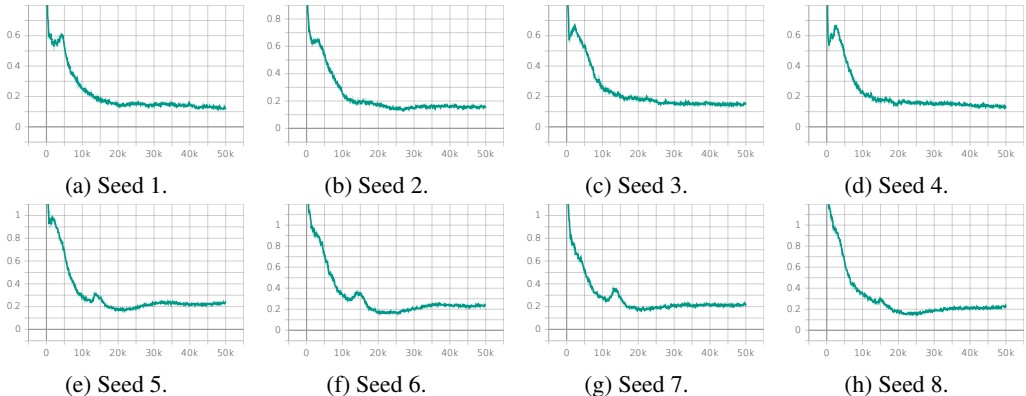

Figure 20: The averaged K̂L of each agent in *Rover-Tower* environments.

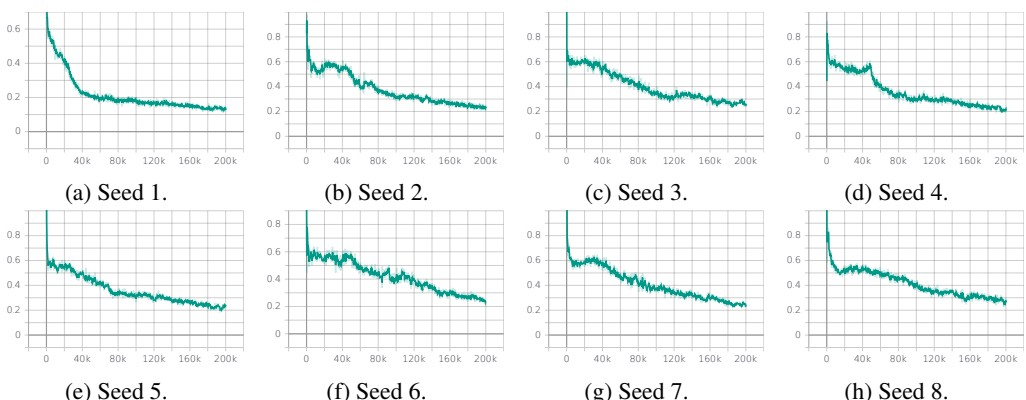

Figure 21: The averaged K̂L of each agent in *Pursuit* environments.

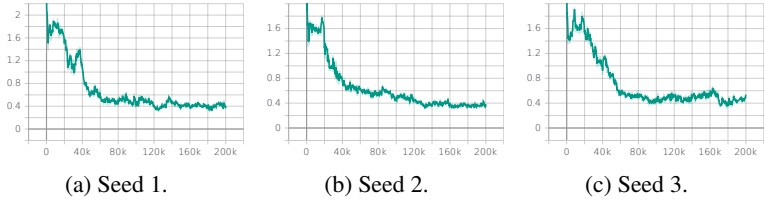

Figure 22: The averaged K̂L of each agent in *Multi-Walker* environments.

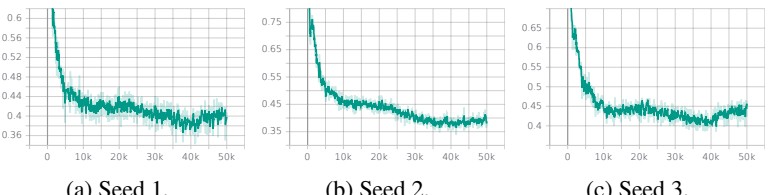

Figure 23: The averaged K̂L of each agent in *Spread* environments.

## REFERENCES FOR SUPPLEMENTARY MATERIAL

Maruan Al-Shedivat, Trapit Bansal, Yura Burda, Ilya Sutskever, Igor Mordatch, and Pieter Abbeel. Continuous adaptation via meta-learning in nonstationary and competitive environments. In *ICLR*, 2018.

Bowen Baker, Ingmar Kanitscheider, Todor Markov, Yi Wu, Glenn Powell, Bob McGrew, and Igor Mordatch. Emergent tool use from multi-agent autocurricula. In *ICLR*, 2019.

Amir Beck and Marc Teboulle. Mirror descent and nonlinear projected subgradient methods for convex optimization. *Operations Research Letters*, 31(3):167–175, 2003.

Omar Besbes, Yonatan Gur, and Assaf Zeevi. Stochastic multi-armed-bandit problem with non-stationary rewards. In *NeurIPS*, 2014.

Wang Chi Cheung, David Simchi-Levi, and Ruihao Zhu. Non-stationary reinforcement learning: The blessing of (more) optimism. *Machine Learning eJournal*, 2019.

Christian Schroeder de Witt, Tarun Gupta, Denys Makoviichuk, Viktor Makoviychuk, Philip HS Torr, Mingfei Sun, and Shimon Whiteson. Is independent learning all you need in the starcraft multi-agent challenge? *arXiv preprint arXiv:2011.09533*, 2020.

Omar Darwiche Domingues, Pierre Ménard, Matteo Pirotta, Emilie Kaufmann, and Michal Valko. A kernel-based approach to non-stationary reinforcement learning in metric spaces. In *International Conference on Artificial Intelligence and Statistics*, pp. 3538–3546, 2021.

Chelsea Finn, Pieter Abbeel, and Sergey Levine. Model-agnostic meta-learning for fast adaptation of deep networks. In *ICML*, 2017.

Jakob Foerster, Richard Y Chen, Maruan Al-Shedivat, Shimon Whiteson, Pieter Abbeel, and Igor Mordatch. Learning with opponent-learning awareness. In *AAMAS*, 2018a.

Jakob N. Foerster, Nantas Nardelli, Gregory Farquhar, Triantafyllos Afouras, Philip H. S. Torr, Pushmeet Kohli, and Shimon Whiteson. Stabilising experience replay for deep multi-agent reinforcement learning. In *ICML*, 2017.

Jakob N Foerster, Gregory Farquhar, Triantafyllos Afouras, Nantas Nardelli, and Shimon Whiteson. Counterfactual multi-agent policy gradients. In *AAAI*, 2018b.

Pratik Gajane, Ronald Ortner, and Peter Auer. A sliding-window algorithm for Markov decision processes with arbitrarily changing rewards and transitions. *arXiv preprint arXiv:1805.10066*, 2018.

Matthieu Geist, Bruno Scherrer, and Olivier Pietquin. A theory of regularized markov decision processes. In *ICML*, 2019.

Jayesh K Gupta, Maxim Egorov, and Mykel Kochenderfer. Cooperative multi-agent control using deep reinforcement learning. In *AAMAS*, 2017.

Tuomas Haarnoja, Aurick Zhou, Pieter Abbeel, and Sergey Levine. Soft actor-critic: Off-policy maximum entropy deep reinforcement learning with a stochastic actor. In *ICML*, 2018.

Eric A Hansen, Daniel S Bernstein, and Shlomo Zilberstein. Dynamic programming for partially observable stochastic games. In *AAAI*, 2004.

Siyue Hu and Jian Hu. Noisy-MAPPO: Noisy advantage values for cooperative multi-agent actor-critic methods. *arXiv preprint arXiv:2106.14334*, 2021.

Shariq Iqbal and Fei Sha. Actor-attention-critic for multi-agent reinforcement learning. In *ICML*, 2019.

Max Jaderberg, Wojciech M. Czarnecki, Iain Dunning, Luke Marris, Guy Lever, Antonio García Castañeda, Charlie Beattie, Neil C. Rabinowitz, Ari S. Morcos, Avraham Ruderman, Nicolas Sonnerat, Tim Green, Louise Deason, Joel Z. Leibo, David Silver, Demis Hassabis, Koray Kavukcuoglu, and Thore Graepel. Human-level performance in 3D multiplayer games with population-based reinforcement learning. *Science*, 364:859 – 865, 2019.

Thomas Jaksch, Ronald Ortner, and Peter Auer. Near-optimal regret bounds for reinforcement learning. *Journal of Machine Learning Research*, 11(51):1563–1600, 2010.

Jiechuan Jiang and Zongqing Lu. Adaptive learning rates for multi-agent reinforcement learning, 2021. URL https://openreview.net/forum?id=yN18f9V1Onp.

Sham Kakade and John Langford. Approximately optimal approximate reinforcement learning. In *ICML*, 2002.

Dong Ki Kim, Miao Liu, Matthew D Riemer, Chuangchuang Sun, Marwa Abdulhai, Golnaz Habibi, Sebastian Lopez-Cot, Gerald Tesauro, and Jonathan How. A policy gradient algorithm for learning to learn in multiagent reinforcement learning. In *ICML*, 2021.

Jakub Grudzien Kuba, Ruiqing Chen, Munning Wen, Ying Wen, Fanglei Sun, Jun Wang, and Yaodong Yang. Trust region policy optimisation in multi-agent reinforcement learning. *arXiv preprint arXiv:2109.11251*, 2021.

Alistair Letcher, Jakob Foerster, David Balduzzi, Tim Rocktäschel, and Shimon Whiteson. Stable opponent shaping in differentiable games. In *ICLR*, 2019.

Hepeng Li and Haibo He. Multi-agent trust region policy optimization. *arXiv preprint arXiv:2010.07916*, 2020.

Timothy P Lillicrap, Jonathan J Hunt, Alexander Pritzel, Nicolas Heess, Tom Erez, Yuval Tassa, David Silver, and Daan Wierstra. Continuous control with deep reinforcement learning. In *ICLR*, 2016.

Siqi Liu, Guy Lever, Josh Merel, Saran Tunyasuvunakool, Nicolas Heess, and Thore Graepel. Emergent coordination through competition. In *ICLR*, 2019.

Ryan Lowe, Yi Wu, Aviv Tamar, Jean Harb, OpenAI Pieter Abbeel, and Igor Mordatch. Multi-agent actor-critic for mixed cooperative-competitive environments. In *NeurIPS*, 2017.

Weichao Mao, Kaiqing Zhang, Ruihao Zhu, David Simchi-Levi, and Tamer Başar. Near-optimal model-free reinforcement learning in non-stationary episodic mdps. In *ICML*, 2021.

Ofir Nachum, Mohammad Norouzi, Kelvin Xu, and Dale Schuurmans. Trust-pcl: An off-policy trust region method for continuous control. In *ICLR*, 2018.

Ronald Ortner, Pratik Gajane, and Peter Auer. Variational regret bounds for reinforcement learning. In *UAI*, 2020.

Georgios Papoudakis, Filippos Christianos, A. Rahman, and Stefano V. Albrecht. Dealing with non-stationarity in multi-agent deep reinforcement learning. *ArXiv*, abs/1906.04737, 2019.

Neil Rabinowitz, Frank Perbet, Francis Song, Chiyuan Zhang, SM Ali Eslami, and Matthew Botvinick. Machine theory of mind. In *ICML*, 2018.

Roberta Raileanu, Emily Denton, Arthur Szlam, and Rob Fergus. Modeling others using oneself in multi-agent reinforcement learning. In *ICML*, 2018.

John Schulman, Sergey Levine, Pieter Abbeel, Michael Jordan, and Philipp Moritz. Trust region policy optimization. In *ICML*, 2015.

John Schulman, Filip Wolski, Prafulla Dhariwal, Alec Radford, and Oleg Klimov. Proximal policy optimization algorithms. *arXiv preprint arXiv:1707.06347*, 2017.

Lior Shani, Yonathan Efroni, and Shie Mannor. Adaptive trust region policy optimization: Global convergence and faster rates for regularized mdps. In *AAAI*, 2020.

Yuhang Song, Jianyi Wang, Thomas Lukasiewicz, Zhenghua Xu, Mai Xu, Zihan Ding, and Lianlong Wu. Arena: A general evaluation platform and building toolkit for multi-agent intelligence. In *AAAI*, 2020.

Justin K Terry, Benjamin Black, Mario Jayakumar, Ananth Hari, Luis Santos, Clemens Dieffendahl, Niall L Williams, Yashas Lokesh, Ryan Sullivan, Caroline Horsch, and Praveen Ravi. PettingZoo: Gym for multi-agent reinforcement learning. *arXiv preprint arXiv:2009.14471*, 2020a.

Justin K Terry, Nathaniel Grammel, Ananth Hari, Luis Santos, and Benjamin Black. Revisiting parameter sharing in multi-agent deep reinforcement learning. *arXiv preprint arXiv:2005.13625*, 2020b.

Gerald Tesauro. Temporal difference learning and TD-Gammon. *Communications of the ACM*, 38 (3):58–68, 1995.

Manan Tomar, Lior Shani, Yonathan Efroni, and Mohammad Ghavamzadeh. Mirror descent policy optimization. *arXiv preprint arXiv:2005.09814*, 2020.

Ying Wen, Hui Chen, Yaodong Yang, Zheng Tian, Minne Li, Xu Chen, and Jun Wang. A game-theoretic approach to multi-agent trust region optimization. *arXiv preprint arXiv:2106.06828*, 2021.

Ronald J Williams. Simple statistical gradient-following algorithms for connectionist reinforcement learning. *Machine Learning*, 8(3-4):229–256, 1992.

Yuhuai Wu, Elman Mansimov, Roger B Grosse, Shun Liao, and Jimmy Ba. Scalable trust-region method for deep reinforcement learning using kronecker-factored approximation. In *NeurIPS*, 2017.

Annie Xie, Dylan Losey, Ryan Tolsma, Chelsea Finn, and Dorsa Sadigh. Learning latent representations to influence multi-agent interaction. In *CoRL*, 2020.

Chao Yu, Akash Velu, Eugene Vinitsky, Yu Wang, Alexandre M. Bayen, and Yi Wu. The surprising effectiveness of MAPPO in cooperative, multi-agent games. *ArXiv*, abs/2103.01955, 2021.

