# OpenReview forum: "Dealing with Non-Stationarity in MARL via Trust-Region Decomposition"
_ICLR.cc/2022/Conference — ICLR 2022 Poster_

### Official Review · Reviewer_16B5 · 2021-11-01

**Correctness:** 3
**Technical Novelty And Significance:** 2
**Empirical Novelty And Significance:** 3
**Recommendation:** 6
**Confidence:** 3

**Main Review:**

Strengths: Dealing with non-stationary in MARL is an important research topic in the community. The proposed method in this paper seems to achieve significant performance improvement than baselines.

Weakness:
The novelty of $\delta$-stationarity is relatively less significant to me, especially when provided the mean-field approximation assumption.  It is suggested that the authors provide more discussion about this.
The writing/structure of this paper needs to be improved. For example, the authors didn't summarise the proposed algorithm. The readers need to spend a hard time trying to combine all proposed ideas to imagine the final algorithm and hence it's difficult to follow. It is suggested that the authors clearly present their algorithm in a compact way.

**Summary Of The Paper:**

This paper studies non-stationarity in MARL. They introduce the notion of $\delta$-stationarity as a measurement of the non-stationarity of a policy sequence. They propose a trust region decomposition framework to impose trust region constraints. They estimate the joint policy divergence with a trust-region decomposition network (TRD-Net), which combined with the mirror descent policy algorithm gives the proposed MAMT algorithm. They show performance improvement in experiments than baselines.

**Summary Of The Review:**

Although the research topic of this paper is important, some parts of the paper should be further discussed and re-organized before it's ready to get accepted.

Update after reading the authors' response: I appreciate the authors' response and revision (especially the algorithm section) that have addressed my concerns. I'm happy to increase my score accordingly.

---

> ### Author Response · Authors · 2021-11-17
> **Response to Official Review of Paper2975 by Reviewer 16B5**
>
> > The novelty of $\delta$-stationarity is relatively less significant to me, especially when provided the mean-field approximation assumption. It is suggested that the authors provide more discussion about this.
>
>
>
> Thank you for your suggestion. We put the discussion with novelty in the common response. Here, we want to discuss more the mean-field assumption. We want to clarify that the mean-field assumption is not introduced in the definition of $\delta$-stationarity, related lemmas, and theorems. We only introduced the mean-field assumption in the algorithm and verified the limitations, which would lead to a severe trust-region decomposition dilemma. Therefore, in implementing MAMT, we introduced technologies such as the pairwise Markov random field, counterfactual baseline, and social influence to make up for a series of problems caused by the mean-field approximation of the joint policy. In addition, the mean-field assumption is widely used in most MARL algorithms based on the CCDA or CTDE framework. Therefore, we believe that the mean-field assumption will not negatively affect the novelty of the $\delta$-stationarity and the MAMT algorithm.
>
>
>
> > The writing/structure of this paper needs to be improved. For example, the authors didn't summarise the proposed algorithm. The readers need to spend a hard time combining all proposed ideas to imagine the final algorithm, and hence it's difficult to follow. It is suggested that the authors clearly present their algorithm in a compact way.
>
>
>
> Thank you very much for your suggestions, and sorry for the inconvenience in your understanding caused by our improper organization of the content. We have reorganized the algorithm section and put a concise version of the algorithm pseudo-code in the main body. The reorganized algorithm part is expanded according to the following logic.
>
> *Solving the trust-region decomposition dilemma need to accurately model the relationship between the local policy divergences and the joint policy divergence. However, calculating the joint policy divergence is intractable. MAMT solves this dilemma from another perspective, i.e., directly learning the relationship through three steps. First, approximating the joint policy divergence; Then, using a differentiable function, i.e., the trust-region decomposition network (TRD-Net), to fit the relationship between the approximated joint policy divergence and local trust-regions; Finally, adaptively adjusting local trust-regions by optimizing the approximated joint policy divergence.*
>
> The content of the first three subsections of the algorithm section is reorganized according to these three steps. After decomposing the joint trust region, the last subsection introduces learning the local policies, summarizes the overall algorithm, and gives the algorithm pseudo-code.

---

> ### Author Response · Authors · 2021-11-30
> **Thanks for raising the score!**
>
> We appreciate the reviewer for raising the score to 6! Thanks for the valuable comments and suggestions!

---

### Official Review · Reviewer_LytU · 2021-11-01

**Correctness:** 4
**Technical Novelty And Significance:** 3
**Empirical Novelty And Significance:** 2
**Recommendation:** 6
**Confidence:** 2

**Main Review:**

The paper is well-written in general and the contributions and the model are well-motivated. Trust region constraints is first mentioned in the abstract, but there has not been enough context provided on how trust region constraints are related to the problem of interest in this paper. That would be great if the authors could briefly elaborate on this in the abstract.

The main algorithm, MAMT, is not presented in the main body of the paper and I think it would be more instructive if the authors could move it from the appendix to the main body.

Is the algorithm a cooperative one? i.e., do agents communicate messages with each other? If so, how does the communication scheme work and what is the communication cost of the proposed algorithm, i.e., how many number of message bits are being communicated through the agents. Communication cost is an important aspect of every  cooperative multi-agent algorithm and a discussion on it and the ways to improve the tradeoff between the dynamic regret and communication cost is appreciated and would benefit the reader.

Could the authors provide a convincing argument about the non-incremental nature of the novelty?

**Summary Of The Paper:**

This paper studies non-stationarity in multi-agent RL. It introduces \delta-stationarity a notion that measures the non-stationarity of a policy sequence. It proposes an algorithm that satisfy \delta-stationarity by approximately constraining the consecutive joint policies’ divergence. A dynamic regret bound is proved for each agent. Experiments that verify the provided theoretical guarantees are provided.

**Summary Of The Review:**

The problem studied in this paper is interesting and I think it would interest the ICLR community. While the theoretical derivations rely on previous works, e.g., Bai et al., 2019; Gao et al., 2021, they are interesting and elegant. I have not checked all the proofs, but they seem correct.

---

> ### Author Response · Authors · 2021-11-17
> **Response to Official Review of Paper2975 by Reviewer LytU [1/2]**
>
> > The paper is well-written in general, and the contributions and the model are well-motivated. Trust region constraints are first mentioned in the abstract. Still, there has not been enough context provided on how trust-region constraints are related to the problem of interest in this paper. That would be great if the authors could briefly elaborate on this in the abstract.
>
>
>
> Thank you very much for your suggestion. In the abstract, our introduction of trust-region constraints is indeed a bit abrupt. In the revised version, we made the following changes:
>
> *A straightforward but highly non-trivial way is to control the joint policies' divergence, which is difficult to estimate accurately by imposing the trust-region constraint on the joint policy.*
>
>
>
> > The main algorithm, MAMT, is not presented in the paper's main body, and I think it would be more instructive if the authors could move it from the appendix to the main body.
>
>
>
> Thank you very much for your suggestions. We have reorganized the algorithm section and put a concise version of the pseudo-code in the main body.

---

> > ### Author Response · Authors · 2021-11-17
> > **Response to Official Review of Paper2975 by Reviewer LytU [2/2]**
> >
> > > Is the algorithm a cooperative one? i.e., do agents communicate messages with each other? How do the communication scheme work and the communication cost of the proposed algorithm, i.e., how many message bits are being communicated through the agents? Communication cost is an essential aspect of every cooperative multi-agent algorithm. A discussion on how to improve the trade-off between the dynamic regret and communication cost is appreciated and would benefit the reader.
> >
> >
> >
> > This is an excellent question, and our algorithm is a cooperative multi-agent reinforcement learning algorithm. But because our algorithm uses a centralized critic and decentralized actor, CCDA (or centralized training and decentralized execution, CTDE) framework, there is no explicit communication between agents in the execution phase. There is only a need for communication between agents in the centralized training phase. Still, it does not make much sense to discuss communication costs under the centralized training framework because what is required here is only computing resources.
> >
> > We think that what may be more interesting is how to achieve the trade-off between communication cost and regret bound in the context of fully decentralized training. In the decentralized training setting, each agent is physically isolated and can only exchange information through communication, and computing resources are also limited and different. The regret bound is not only related to the degree of control of non-stationarity but also related to the policy learning itself. In a fully decentralized setting, there has been a lot of research on the latter in recent years. The discussion of these works is beyond the scope of this paper. Here we will mainly discuss the trade-off between the communication cost and the control of non-stationarity under the fully decentralized training setting.
> >
> > In the MAMT algorithm, the trust-region decomposition network (TRD-Net) is involved in communication except for policy learning. TRD-Net is the core graph neural network of the MAMT algorithm. Its primary purpose is to fit the differentiable function relationship between the local trust regions and the approximated consecutive joint policy divergence to constrain the agent by controlling the latter. Once TRD-Net does not converge to a better solution due to communication bandwidth limitations under the decentralized training framework, the non-stationarity \delta is challenging to control well, making the regret bound more relaxed.
> >
> > In the current experimental setting, the communication topology of TRD-Net is a fully connected network, and $128$-dimensional real number vectors must be transmitted to each other in each training iteration. Therefore, when the number of agents is large, MAMT requires a high communication cost. However, in the process of hyperparameter tuning, we found that reducing the dimensions of the transmitted information, for example, to $64$ or $32$ dimensions will cause a specific performance loss, but it is within an acceptable range. Therefore, some dimensionality reduction techniques, self-encoding methods, or the information bottleneck can remove redundant information. And there is also room for optimization in the communication topology. For example, it only needs to communicate with neighboring agents, verified in many MARL algorithms based on GNNs. Finally, in decentralized training, the agent does not need to communicate all the time. Pre-defined or learned rules (learning to communicate) can be used to control when the agent communicates. Therefore, in decentralized training, it is possible to achieve a good trade-off between communication cost and regret bound by appropriately reducing information dimensions, simplifying communication topology, and reducing communication frequency.
> >
> >
> >
> > > Could the authors provide a convincing argument about the non-incremental nature of the novelty?
> >
> >
> >
> > Thank you for your advice. We put the discussion with novelty in the common response.

---

### Official Review · Reviewer_DdBz · 2021-11-02

**Correctness:** 3
**Technical Novelty And Significance:** 2
**Empirical Novelty And Significance:** 2
**Recommendation:** 6
**Confidence:** 3

**Main Review:**

**Strength:**
This paper makes a logical and theoretical connection to the proposed method based on the $\delta$-stationarity, its bound, and trust-region decomposition dilemma.

**Weaknesses and Concerns:**
1. I am unsure whether the statement "the insufficient theoretical understanding of non-stationarity" is true. In the game-theoretical MARL literature, the non-stationarity is represented by a distance of joint policy to an equilibrium. For instance, NashConv is a commonly used metric to measure the distance from a Nash equilibrium (Lanctot et al., 2017).

*Marc Lanctot, Vinicius Zambaldi, Audrunas Gruslys, Angeliki Lazaridou, Karl Tuyls, Julien Perolat, David Silver, Thore Graepel. A Unified Game-Theoretic Approach to Multiagent Reinforcement Learning. NeurIPS, 2017.*

2. I am unsure how novel the new metric $\delta$-stationarity is because it essentially measures the maximum KLD between two policies.

3. Regarding the considered baseline methods in Section 4, is a fair comparison to MAMT conducted? For example, MAMT employs GNN layers, which could have a larger representation capacity than fully-connected layers. I wonder whether the baselines have a similar network architecture to MAMT for a fair comparison. As another related concern, MAMT's implementation uses other recent techniques, such as attention, mirror descent, and the counterfactual baseline, so it is difficult to identify whether the improvement over baselines is mainly due to which factor.

4. Regarding writing, the paper motivates and explains the idea for a general multiagent environment (e.g., "opponent" switching cost). However, the proposed method cannot be applied to competitive and/or general-sum settings because an agent cannot directly control the learning of other agents in these settings. In particular, limiting the divergence between an agent's consecutive policies in competitive settings would not be desirable because the opponent can learn faster, exploit the agent's (relatively) slow learning, and thus gain a bigger return than the agent. I also note that experiments are conducted on the four coordination environments. As a result, rewriting the paper assuming cooperative MARL settings can improve the paper's clarity.

5. Regarding related works in A.1.2, other recent works exploit the opponent's learning process to achieve better performance. Adding the following works will make the related work section more complete:
*Alistair Letcher, Jakob Foerster, David Balduzzi, Tim Rocktäschel, Shimon Whiteson. Stable Opponent Shaping in Differentiable Games. ICLR, 2019.*

*Annie Xie, Dylan P. Losey, Ryan Tolsma, Chelsea Finn, Dorsa Sadigh. Learning Latent Representations to Influence Multi-Agent Interaction. CoRL, 2020.*

*Dong-Ki Kim, Miao Liu, Matthew Riemer, Chuangchuang Sun, Marwa Abdulhai, Golnaz Habibi, Sebastian Lopez-Cot, Gerald Tesauro, Jonathan P. How. A Policy Gradient Algorithm for Learning to Learn in Multiagent Reinforcement Learning. ICML, 2021.*

**Summary Of The Paper:**

This paper proposes MAMT to address the important non-stationarity challenge in MARL, which is inherently resulted from the learning of other agents. Specifically, this paper makes two contributions: 1) introduces the $\delta$-stationarity metric to measure the non-stationarity, which bounds the dynamic regret of each agent; and 2) addresses the trust-region decomposition dilemma by combining message passing and mirror descent in MAMT. Experiments in four coordinated domains show that MAMT outperforms prior related works.

**Summary Of The Review:**

I initially vote for a score of 5. While the topic studied in this paper is an important problem in MARL and this paper makes a logical motivation to MAMT, I have main concerns regarding the experimental results (i.e., fair comparisons against baselines and identifying the main source of the improvements) and writing of the paper (i.e., needs to be rewritten for a cooperative multiagent setting). After reading the authors' responses to my questions, I am open to raising my score.

---

> ### Author Response · Authors · 2021-11-17
> **Response to Official Review of Paper2975 by Reviewer DdBz [1/3]**
>
> > I am unsure whether the statement "the insufficient theoretical understanding of non-stationarity" is true. In the game-theoretical MARL literature, the **non-stationarity is represented by a distance of joint policy to an equilibrium**. For instance, NashConv is a commonly used metric to measure the distance from a Nash equilibrium (Lanctot et al., 2017).
> >
> > *Marc Lanctot, Vinicius Zambaldi, Audrunas Gruslys, Angeliki Lazaridou, Karl Tuyls, Julien Perolat, David Silver, Thore Graepel. A Unified Game-Theoretic Approach to Multiagent Reinforcement Learning. NeurIPS, 2017*
>
>
>
> Thank you very much for your suggestion, and sorry that we have improperly written here. We have modified the original statement in the revised version. We want to express that the existing work has comprehensively studied the consequences of non-stationarity and has proposed various indicators to measure the outcomes of non-stationarity, such as dynamic regret, NashConv, etc. However, we believe that these measurements are indirect definitions of non-stationarity. Some are still defined under specific problem settings (for example, NashConv is more applied to games with imperfect information). Still, it lacks the definition of the phenomenon of non-stationarity itself.
>
>
>
> > I am unsure **how novel the new metric** $\delta$-stationarity is because it essentially measures the maximum KLD between two policies.
>
>
>
> Thank you for your question. We put the discussion with novelty in the common response.

---

> > ### Author Response · Authors · 2021-11-17
> > **Response to Official Review of Paper2975 by Reviewer DdBz [2/3]**
> >
> > > Regarding the considered baseline methods in Section 4, is a fair comparison to MAMT conducted? For example, MAMT **employs GNN layers**, which could have a larger representation capacity than fully-connected layers. I wonder whether the baselines have a similar network architecture to MAMT for a fair comparison. As another related concern, MAMT's implementation uses other recent techniques, such as attention, mirror descent, and the counterfactual baseline, so it is difficult to identify whether the improvement over baselines is **mainly due to which factor**.
> >
> >
> >
> > Thank you very much for your question. In comparison with the baseline, we have ensured fairness as much as possible, but the ablation study of MAMT itself is still insufficient. We have added some new comparative experiments to the revised version (due to the time of the experiment, we will add all the results of the experiment to the revised version in the next few days). Below we will explain whether the comparison with baselines is fair and the newly added comparative experiments.
> >
> > - Comparison with baselines.
> >   - **Regarding the use of GNN**. We did not use the GNN architecture in the MAMT policy networks, which is the same as all baselines. In MAMT, GNN is proposed to estimate better the consecutive joint policy divergence to adjust the local trust regions reasonably. The weight of edges in GNN is critical to performance, so we conducted an additional ablation study, which will be described later.
> >   - **Regarding the use of the attention mechanism.** Since the MAAC algorithm is the backbone of MAMT, both use the attention mechanism in the agent's policy network to filter the redundant information generated during centralized training. For a fair comparison, we also introduced the same attention mechanism in the MA-PPO algorithm. Only the MADDPG algorithm does not use the attention mechanism because we hope to verify the effect of the attention mechanism on performance through the MADDPG algorithm. It can be seen from the experiment that in more complex cooperative scenarios, the attention mechanism will have a significant impact on performance.
> > - More ablation studies (due to the space limitation of the main body, we put it in the Appendix. H section.).
> >   - **Regarding the mirror descent.** Considering the low sample efficiency of MARL, we did not adopt the fully on-policy learning but used off-policy training with a small replay buffer. In this way, it can be ensured that samples are less different from the current policy, thereby alleviating the instability caused by off-policy training. The methods such as TRPO and PPO are all on-policy algorithms. Although there are some off-policy trust-region methods, such as Trust-PCL, etc., they are all more complicated. To this end, we adopted a more concise technique, Mirror Descent, as the optimization method of MAMT. MAMT can also use the on-policy method as the backbone, such as PPO. To this end, we have added a comparative experiment, replacing Mirror Descent with the PPO algorithm.
> >   - **Regarding the counterfactual baseline.** In the MAMT algorithm, the counterfactual baseline is used in policy learning and used in the calculation of coordination coefficients. To verify the relationship between the correctness of the modeling of the dependencies between agents and the algorithm's performance, we have added two comparative experiments to the revised version. Specifically, we used two pre-defined ways to set the coordination coefficients, set to a fixed value of $1/(n-1)$ (where $n$ is the number of agents), and random sampling (random sampling from the range of $(0, 1)$ and then input into the $\mathrm{softmax}$ function).
> >   - **Regarding the modeling of others.** We borrowed the idea in offline RL to measure the degree of out-of-distribution (see Equation (5)) through the model-based RL to reasonably characterize the non-stationarity or the consecutive joint policy divergence. However, a more intuitive and simple method is to directly perform a weighted summation of the consecutive local policy divergence (the weights are the coordination coefficients). The reason why we did not adopt this scheme is based on the following considerations. It can be seen from Definition 3 that the agents' policies do not directly cause the non-stationarity. The agent cannot directly observe the opponent's policy but only the action sequences. This explicit information directly affects the learning process of the agent and leads to non-stationarity. To verify the correctness of the idea, we also did additional comparative experiments. Specifically, we changed the calculation method of coordination coefficient to the weighted summation of consecutive local policy divergence, i.e. $1/n\cdot\sum\_i \mathcal{C}\_{i,j} \sum\_{j \neq i} \mathrm{KL}[ \pi^{\prime}\_{\psi\_j}(o\_j^{t}) \| \pi\_{\psi\_j}(o\_j^t)]$.

---

> > > ### Author Response · Authors · 2021-11-17
> > > **Response to Official Review of Paper2975 by Reviewer DdBz [3/3]**
> > >
> > > > Regarding writing, the paper motivates and explains the idea for a general multi-agent environment (e.g., "opponent" switching cost). However, the proposed method cannot be applied to competitive and/or general-sum settings because an agent cannot directly control the learning of other agents in these settings. In particular, limiting the divergence between an agent's consecutive policies in competitive settings would not be desirable because the opponent can learn faster, exploit the agent's (relatively) slow learning, and thus gain a bigger return than the agent. I also note that experiments are conducted on the four coordination environments. As a result, **rewriting the paper assuming cooperative MARL settings** can improve the paper's clarity.
> > >
> > >
> > >
> > > Thank you very much for your suggestion. Your suggestion is undoubtedly correct. Our method belongs to the setting of cooperative MARL. We have made the following additions and changes in the revised version:
> > >
> > > - In the Abstract, the third and fifth paragraphs of Section 1, the second and third paragraphs of Section 2, and the first paragraph of Section 3 have added the modifier of "*cooperative*" to our method;
> > > - We amend the first sentence of the opening as follows: *Learning how to achieve effective collaboration in multi-agent decision-making tasks, such as multi-player games, resource allocation, and network routing, is a significant problem in cooperative multi-agent reinforcement learning (MARL)*.
> > > - We added the first paragraph of Section 2 to formally define the cooperative MARL problem to be solved in this paper.
> > >
> > >
> > >
> > > > Regarding related works in A.1.2, other recent works exploit the opponent's learning process to achieve better performance. **Adding the following works** will make the related work section more complete:
> > > >
> > > > *Alistair Letcher, Jakob Foerster, David Balduzzi, Tim Rocktäschel, Shimon Whiteson. Stable Opponent Shaping in Differentiable Games. ICLR, 2019*
> > > >
> > > > *Annie Xie, Dylan P. Losey, Ryan Tolsma, Chelsea Finn, Dorsa Sadigh. Learning Latent Representations to Influence Multi-Agent Interaction. CoRL, 2020*
> > > >
> > > > *Dong-Ki Kim, Miao Liu, Matthew Riemer, Chuangchuang Sun, Marwa Abdulhai, Golnaz Habibi, Sebastian Lopez-Cot, Gerald Tesauro, Jonathan P. How. A Policy Gradient Algorithm for Learning to Learn in Multiagent Reinforcement Learning. ICML, 2021*
> > >
> > >
> > >
> > > Thank you very much for your pieces of advice. We have added a discussion of the above three papers in the related work section of the revised version. We denote the algorithms proposed in the three papers as SOS, LILI, and Meta-MAPG, and a summary of the three works is as follows.
> > >
> > > Stable Opponent Shaping (SOS) is presented to solve the convergence problem shown by LOLA under certain n-player and non-convex games. This new method interpolates between LOLA and a stable variant named LookAhead, which is proved that converges locally to equilibria and avoids strict saddles in all differentiable games. Unlike LOLA and SOS directly predicting the opponent's policy parameters, LILI solves the non-stationary problem by predicting and influencing the latent representation of the opponent's policy. Recently, Meta-PG transformed non-stationarity problems into meta-learning problems and extended MAML to MAS to find an initialization policy that quickly adapted to non-stationarity. However, Meta-PG treats other agents as external factors whose learning it cannot affect, which does not hold in practice for the general multi-agent learning settings. Meta-MAPG combines LOLA to consider both an agent's own non-stationary policy dynamics and the non-stationary policy dynamics of other agents in the environment. Due to the unique training mechanism of the above methods, they are difficult to extend to the tasks of more than two agents.

---

> > > > ### Comment · Reviewer_DdBz · 2021-11-27
> > > > **Rebuttal Response**
> > > >
> > > > I would like to thank the authors for their detailed responses to my concerns and for making the changes to the paper accordingly. The updated version of the paper addresses some of my main concerns. My recommendation is between 5 and 6, and I have increased my recommendation from 5 to 6. I agree with the other reviewers that the writing/presentation of the paper can be improved further, and I am still concerned that MAMT is somewhat a sophisticated algorithm that needs attention, mirror descent, and counterfactual baseline techniques.

---

> > > > > ### Author Response · Authors · 2021-11-28
> > > > > **Thanks for raising the score to 6!**
> > > > >
> > > > > Thank you very much for your recognition of our work. We will continue to polish the revised version, and we also welcome you to put forward more constructive comments on the writing and presentation of the paper.
> > > > >
> > > > > In addition, regarding the complexity of the proposed algorithm you mentioned, we would like to add a few points. The MAMT algorithm does need to use some technologies in the deep learning community and the MARL community to ensure that the algorithm is effective. The deep learning community widely uses the attention mechanism and the mirror descent optimization method, and their principles are straightforward. They have also been integrated into some deep learning toolkits, such as PyTorch [1] and TensorFlow [2]. Therefore, introducing the attention mechanism and mirror descent optimization method into the algorithm will not bring too much algorithmic complexity.
> > > > >
> > > > > Furthermore, the counterfactual baseline is widely used in the MARL community (such as COMA [3], MAAC [4], MACKRL [5], etc.) to solve the multi-agent credit assignment problem. In addition to using counterfactual baselines for the above purpose, MAMT also uses it to model the dependencies between agents. Therefore, compared with state-of-the-art MARL algorithms, MAMT does not bring extra algorithmic complexity.
> > > > >
> > > > > **References**
> > > > >
> > > > > [1] Paszke, Adam, et al. "Pytorch: An imperative style, high-performance deep learning library." NeurIPS (2019).
> > > > >
> > > > > [2] Abadi, Martín, et al. "Tensorflow: Large-scale machine learning on heterogeneous distributed systems." ArXiv Preprint (2016).
> > > > >
> > > > > [3] Foerster, Jakob, et al. "Counterfactual multi-agent policy gradients." AAAI (2018).
> > > > >
> > > > > [4]  Iqbal, Shariq and Fei Sha. "Actor-Attention-Critic for Multi-Agent Reinforcement Learning." ICML (2019).
> > > > >
> > > > > [5] Foerster, Jakob N. et al. "Multi-Agent Common Knowledge Reinforcement Learning." NeurIPS (2019).

---

> > > > > > ### Comment · Reviewer_DdBz · 2021-11-29
> > > > > > **Rebuttal Response**
> > > > > >
> > > > > > Thank you for the clarification regarding the complexity. Yes, I will provide additional suggestions on the writing and presentation once the paper is accepted.

---

> > > ### Author Response · Authors · 2021-11-20
> > > **Ablation Studies Updated (Revision Uploaded)**
> > >
> > > The results of all new ablation studies have been updated in Appendix H of the revised version.

---

### Official Review · Reviewer_XYrN · 2021-11-05

**Correctness:** 3
**Technical Novelty And Significance:** 3
**Empirical Novelty And Significance:** 3
**Recommendation:** 8
**Confidence:** 4

**Main Review:**


Strengths:
- I like the idea of controlling the non-stationarity of the policies to guide learning in the MARL setting.
- The proposed switching cost seems to be a useful metric to bound the non-stationarity of the agents' policies.
- Although it is quite natural that the mean-field approximation directly decomposes the overall switching cost into individual switching costs, I found the connection to be quite interesting.
- Finally, the experimental section demonstrates the effectiveness of the proposed approach compared to the other algorithms, and also it is an interesting results that the non-stationarity gradually reduces as the number of training rounds is increases.

Weaknesses:
- I found the paper hard to follow at times. Particularly, the introduction and the related work assumes significant background knowledge.
- The particular measure of non-stationarity $\mathcal{D}^t_{i,ns}$, as defined in equation (5), is not well-motivated. The authors should have provided some reasons behind choosing this function.

Some questions for the authors.
1. The bound on theorem 1 has a linear dependence on the number of time intervals $T = HK$. Therefore, the average regret doesn't vanish to zero as T becomes large, even when non-stationarity is small. What is the reason for this linear dependence?
2. What is the justification behind replacing the $\delta_i$ in equation (3) by $\delta/n$ in equation (4)? Probably an alternative design choice is to keep the constraint $\sum_i \delta_i \le \delta$ and do a projection after each step of the gradient descent.

Some comments about the presentation.
1. I found the related work section hard to follow. However, this could be because I am not familiar with the latest work on MARL. But the authors should formally introduce MDP, POMDP, POSG through notations before discussing them in the related section. This could make the related section easy to follow.
2. Some notations should have been formally defined in the paper e.g. $\Pi_{ns}, POSG$ etc.

=============Update after reading the authors' response================
Many thanks for your response. I am happy with the way you addressed my two main concerns and I would vote for accepting the paper.

**Summary Of The Paper:**

This paper proposes a new learning algorithm for multi-agent reinforcement learning (MARL) when the stationarity of the policies are bounded. Non-stationarity measures how the agents' policies change with time and high non-stationarity implies slow learning in an MARL setting. The authors introduced a measure based on the KL-divergence between any pair of opponent's joint policies. The first result states that if the switching cost is bounded from above, then it is possible to derive a regret on the learning algorithm used by each agent.

The main problem is to maximize the sum of values of the agents subject to a bounded switching cost. The authors consider a mean-field approximation which assumes that the joint policy factorizes into individual policies conditioned on the observation. Based on this observations, they are able to show that it is enough to bound the switching cost of each agent rather than the overall switching cost. The authors propose a bilevel optimization problem where the outer optimization is over $\delta$ the measure of switching cost and the inner variable $\theta^\star(\delta)$ is the optimal parameter for a given choice of $\delta$. The proposed optimization procedure alternates between gradient descent at different scales.



**Summary Of The Review:**

Overall, I think the paper makes a useful contribution to the vast literature on MARL. In particular, I like the idea of using switching cost to derive better learning algorithms. The experimental evaluation also demonstrates the benefits of the proposes approach. That being said, I thought that the paper should do a better job in presenting the main ideas. Additionally, I believe bounds of some theorems (e.g. theorem 1) can be improved.

---

> ### Author Response · Authors · 2021-11-17
> **Response to Official Review of Paper2975 by Reviewer XYrN [1/2]**
>
> > I found the paper hard to follow at times. Particularly, the **introduction** and the **related work** assume significant background knowledge. The authors should formally introduce MDP, POMDP, POSG through notations before discussing them in the related section. This could make the related section easy to follow. And some **notations should have been formally defined** in the paper, e.g., $\Pi_{n s}$, $POSG$, etc.
>
>
>
> Thank you very much for your suggestions, and sorry for the inconvenience in your understanding caused by our improper organization of the content. We have made the following changes to the paper in the revised version:
>
> - In the first paragraph of the introduction, when describing non-stationary problems, we no longer use formal descriptions and remove relevant terms and mathematical symbols, such as $POSG$, $\mathcal{T}_i$, $\mathcal{R}_i$.
> - We added the formal definition of POSG at the beginning of the second part of the paper and the definition of the problem (i.e., cooperative tasks) that this paper wants to solve. And the definition of $\Pi_{n s}$ is also added in Section 3.2.
> - In the appendix, we have changed the order of the related work and preliminary two parts to facilitate understanding.
>
>
>
> > The particular **measure of non-stationarity** $\\mathcal{D}\_{i, n s^{\\prime}}^{t}$ as defined in equation (5), is not well-motivated. The authors should have provided some reasons behind choosing this function.
> > $$
> > \\mathcal{D}\_{i, \\mathrm{~ns}}^{t}=\\Pi_{\\mathrm{ns}}\\left(\\sum\_{j \\neq i} \\mathcal{C}\_{i, j}^{t}\\left(\\mathrm{KL}\\left[h_{\\phi_{i}^{j}}\\left(o_{i}^{\\iota}\\right) \\| \\pi_{\\psi\_{j}}\\left(o\_{j}^{\\ell}\\right)\\right]\\right)\\right) \\nonumber \\quad (5)
> > $$
>
>
> Thanks for your advice. We have given the motivation in the revised version, which is now organized as follows. From the analysis of the trust-region decomposition dilemma, it can be seen that the dependence between agents is an essential factor affecting the joint policy divergence. According to Theorem 2, Definition 2 and Assumption 1, after modeling the coordination coefficient between agents, does it mean that the weighted summation of the local policy divergence, i.e., $1/n\\cdot\\sum\_i \\mathcal{C}\_{i,j} \\sum\_{j \\neq i} \\mathrm{KL}[ \\pi^{\\prime}\_{\\psi\_j}(o\_j^{t}) \\| \\pi\_{\\psi\_j}(o\_j^t)]$ can accurately estimate the joint strategy divergence? It can be seen from Definition 3 that the agents' policies do not directly cause the non-stationarity. The agent cannot directly observe the opponent's policy but only the action sequences. This explicit information directly affects the agent's learning process and leads to non-stationarity (to verify the correctness of the idea, we also did additional comparative experiments. Specifically, we changed the calculation method of coordination coefficient to the weighted summation of consecutive local policy divergence).
>
> Therefore, we borrowed the idea in offline RL to measure the degree of out-of-distribution through the model-based RL to reasonably characterize the non-stationarity. Specifically, each agent $i$ has a prediction model $h_{\phi_i^j}$ for each other agent $j$, predicting the other agent's actions based on its local history observation. The prediction model can naturally replace the old policy $\pi^{\prime}$ since it is trained using historical information. Thus the summation of the divergence between the predicted action distribution and the actual action distribution of all other agents could represent the non-stationarity of the agent $i$, i.e.,
>
> $$
> \\begin{equation} \\mathcal{D}\_{i,\\mathrm{ns}}^t = \\Pi\_{\\mathrm{ns}} \\left( \\textstyle{\\sum}\_{j \\neq i}\\mathcal{C}\_{i,j}^t\\left(\\mathrm{KL}\\left[ h\_{\\phi^j\_i}(o\_i^t) \\| \\pi\_{\\psi\_j}(o\_j^t)\\right]\\right)\\right), \\end{equation}
> $$
>
> where “ns" denotes the ”non-stationarity"; $\\pi\_{\\psi\_j}$ is the policy of agent $j$ which is parameterized by $\\psi\_j$; $\\Pi\_{\\mathrm{ns}}$ represents the projection function, which constrains $\\mathcal{D}\_{i,\\mathrm{ns}}^t$ to a specific range to stabilize the training procedure. The joint policy divergence can then be approximated by the non-stationarity of MAS, i.e., the summation of all local non-stationarities, $\\mathcal{D}\_{\\mathrm{ns}}^t = \\sum\_i \\mathcal{D}\_{i,\\mathrm{ns}}^t$.

---

> > ### Author Response · Authors · 2021-11-17
> > **Response to Official Review of Paper2975 by Reviewer XYrN [2/2]**
> >
> > > The bound on theorem 1 has a linear dependence on the number of time intervals $T=H K$. Therefore, the average regret doesn't vanish to zero as $T$ becomes large, even when non-stationarity is small. What is **the reason for this linear dependence**?
> > >
> >
> >
> >
> > This is an excellent question. Due to space limitations, we have not discussed it in the current version, so we have added a section (Appendix. F) to the supplementary material of the revised version to discuss this issue. The core point we want to express in this part is, although theoretically only linear regret can be achieved, it can be close to sub-linear in actual implementation. In general, constraining $\delta$-stationarity produces a linear regret because the opponent switching cost is a relatively loose constraint. The key to achieving sub-linear regret is to constrain the consecutive joint policy divergence at each timestep, not the maximum or expected value of the entire learning procedure.
> >
> > Therefore, we propose a tighter temporal constraint $\delta^t$ on the basis of opponent switching cost, and prove that by imposing this constraint on the consecutive joint policy divergence at each timestep, the algorithm can achieve sub-linear regret $\textstyle{\tilde{O}(D_{\max}^{3/2}|\mathcal{O}|^{1/2}\Delta_{\delta}^{1/4}T^{1/2})}$, in the supplementary material. Although MAMT has a global hyperparameter $\delta$ to constrain the adaptively adjusted $\{\delta_i\}_{i=1}^N$, the constraint only works when $\sum_i \delta_i > \delta$. We have observed in the experiment that most of the time $\sum_i \delta_i \leq \delta$ (see Table 2 in the Appendix. F of the revised version). In this way, there is a different $\delta^t=\sum_i \delta^t_i$ at each timestep $t$ to constrain the consecutive joint policy divergence, thereby approximately achieving a sub-linear regret.
> >
> >
> >
> > > What is the justification behind **replacing the $\delta_{i}$ in equation (3) by $\delta / n$ in equation (4)**? Probably an alternative design choice is to keep the constraint $\sum_{i} \delta_{i} \leq \delta$ and do a projection after each step of the gradient descent.
> > >
> > > $$
> > > \mathbb{E}\_{o\_{i} \sim \mathcal{D}}\left[\mathbb{E}\_{a\_{i} \sim \pi\_{i}}\left[D\_{\mathrm{KL}}\left(\pi\_{i}\left(a\_{i} \mid o\_{i}\right), \pi\_{i}^{k}\left(a\_{i} \mid o\_{i}\right)\right)\right]\right] \leq \delta\_{i}, \quad \forall i \nonumber \quad (3)
> > > $$
> > >
> > > $$
> > > \pi^{k+1} \in \arg \max \sum\_{i=1}^{n} V\_{i}^{\pi}(o), \forall o, \text { s.t. } \mathbb{E}\_{o\_{i} \sim \mathcal{D}}\left[\mathbb{E}\_{a\_{i} \sim \pi\_{i}}\left[\operatorname{KL}\left(\pi\_{i}\left(a\_{i} \mid o\_{i}\right), \pi\_{i}^{k}\left(a\_{i} \mid o\_{i}\right)\right)\right]\right] \leq \delta / n, \forall i \nonumber \quad (4)
> > > $$
> > >
> >
> >
> >
> > This is a detailed question, and I'm sorry we didn't elaborate in the paper. (4) is mainly used to elicit the existence of the trust-region decomposition dilemma and to compare with our algorithm as a baseline in the experimental part. We did not use the decomposition method in (4) in the implementation of the MAMT algorithm. From the experimental results, although the consecutive joint policy divergence can be constrained concisely by (4), using (4) will significantly degrade performance due to the trust-region decomposition dilemma.
> >
> > In implementing MAMT, we have adopted a method similar to what you said (projection) to meet the constraints. Specifically, we added an additional regular term $\mathrm{ReLU}(\sum_i \delta_i - \delta)$ to $\mathcal{F}(\boldsymbol{\delta})$ so that $\sum_i\delta_i$ can satisfy the constraints. This regular term guarantees that it will be punished once the constraint is violated, that is, $\sum_i\delta_i > \delta$. We added the above implementation details to the revised version at the same time.

---

> ### Author Response · Authors · 2021-11-30
> **Thanks for raising the score to 8**
>
> We really appreciate the feedback from the reviewer and thanks for raising the score!

---

### Author Response · Authors · 2021-11-17
**Common Response (Revision Uploaded) [1/2]**

We thank all reviewers for their valuable comments. Based on all reviewers' comments, we have summarized two main issues, which will be discussed in the common response, and we have responded to the remaining problems separately. The two main questions are the novelty of $\delta$-stationarity and MAMT algorithm, and the organization of the paper's content, primarily related work and algorithm. Below we discuss these issues separately.

- **Regarding the novelty of $\delta$-stationarity and MAMT algorithm.** We will analyze novelty from the definition of $\delta$-stationarity and algorithm design.

  - **The $\delta$-stationarity.** Existing work has fully studied the consequences of non-stationarity and has proposed various indicators to measure the consequences of non-stationarity, such as dynamic regret, NashConv, etc. However, we believe that these measurements are indirect definitions of non-stationarity. Some are still defined under specific problem settings (for example, NashConv is more applied to games with imperfect information). Still, it lacks the definition of the phenomenon of non-stationarity itself. The non-stationarity problem in multi-agent reinforcement learning has been discussed and studied in many works of literature from the early to the present and is generally informally defined as follows: *The non-stationarity is caused by changing agents' policies during the learning process.* This is a very concise statement, but it summarizes the key factors that lead to non-stationary generation. We believe that such a concise statement should also correspond to a concise mathematical form because a more sophisticated mathematical formulation means worse generalization.

    Therefore, based on the definition of the non-stationary MDP algorithm in the single-agent RL community, we present $\delta$-stationarity to define the non-stationarity of MARL. $\delta$-stationarity actually measures the KL divergence of consecutive joint policies, but this definition is concise enough to apply to any problem setting. Furthermore, we also proved the relationship between $\delta$-stationarity and the dynamic regret bound, which is widely used to measure the consequences of non-stationarity. This also verifies the rationality of $\delta$-stationariy. We believe that $\delta$-stationarity can also be linked to more indicators that measure the outcomes of non-stationarity.

    In addition, $\delta$-stationarity can also inspire concise but effective algorithms by virtue of its concise characteristics. The motivation of the MAMT algorithm is very straightforward. Because of the proportional relationship between $\delta$-stationarity and consecutive joint policy divergence, we can directly constrain the latter to alleviate the non-stationarity. Of course, in order to more accurately estimate the consecutive joint policy divergence and be more tractable to solve the optimization problem, we have added other modules to the MAMT algorithm. Experimental results point that MAMT does show a better effect in dealing with the non-stationary problem in cooperative tasks. We hope that the simplicity of $\delta$-stationarity can inspire more simple yet efficient MARL algorithms and more evaluation indicators to measure the consequences of non-stationarity.

---

> ### Author Response · Authors · 2021-11-17
> **Common Response [2/2]**
>
> - **Regarding the novelty of $\delta$-stationarity and MAMT algorithm.** We will analyze novelty from the definition of $\delta$-stationarity and algorithm design.
>   - **The algorithm design.** Firstly, from the analysis of the trust-region decomposition dilemma, it can be seen that the dependence between agents is an essential factor affecting the joint policy divergence. Thus, we introduce a non-negative real number, *coordination coefficient*, to represent the dependency between two agents. Specifically, we combine the counterfactual baseline with social influence and propose a novel method of modeling agent dependencies. This method only relies on the central critic of the agent and can be applied to any CCDA or CTDE framework.
>
>     Secondly, we propose a novel way to approximate the KL divergence of the agent's consecutive joint policies. It can be seen from the definition of $\delta$-stationarity that the agents' policies do not directly cause the non-stationarity. The agent cannot directly observe the opponent's policy but only the action sequences. This explicit information directly affects the learning process of the agent and leads to non-stationarity. Therefore, we borrow from the idea of measuring the degree of out-of-distribution through model-based methods in offline RL to estimate the consecutive joint policy divergence instead of performing a weighted (weights are the cooperation coefficients) summation of the local policy divergence.
>
>     Thirdly, we propose a novel network structure to model the relationship between the local trust regions and the approximated consecutive joint policy divergence. Recent works modeled the joint policy as a Markov random field with pairwise interactions based on graph neural networks (GNN) to achieve a more reasonable joint policy decomposition. The consecutive joint policy divergence is related to the local policies and local trust regions. Inspired by these methods, a similar mechanism is utilized in this paper to decompose the consecutive joint policy divergence (rather than the joint policy itself) into local policies and local trust regions (which can be input to the GNN) with pairwise interactions.
>
>     In general, the various techniques used in the design of the MAMT algorithm have achieved good results in their respective problems. Still, they are officially used to solve the non-stationary problem in MARL for the first time. This paper combines these methods organically and has achieved visible results in solving non-stationary problems.
>
> - **Regarding the organization of the content of the paper.** We have made the following changes to the paper (the main body and the supplementary material, the modified parts are marked in red):
>
>   - **Main body**
>     - We added the assumption of cooperative MARL in the first and second sections.
>     - We replaced the mathematical symbols in the introduction section with descriptive language to improve the paper's readability.
>     - We formally modeled the cooperative MARL problem that this paper wants to solve and added a brief definition of POSG in the second section.
>     - We reorganized the algorithm part of the paper into a more compact form. We moved the pseudo-code of the MAMT algorithm in the supplementary material to the main body so that readers can have a clearer overall understanding of the MAMT algorithm.
>   - **Supplementary material**
>     - We moved the preliminaries before the related work to improve the readability of the paper.
>     - We have added some recent representative work to the related work section to increase the completeness of this section.
>     - We have added a section (Appendix. F) to discuss the linear regret bound in Theorem 1 in more depth.
>     - We have added additional ablation studies  (Appendix. H) to better support the various ideas presented in this paper.

---

> > ### Author Response · Authors · 2021-11-28
> > **Status update**
> >
> > We would like to thank Reviewer DdBz for raising the score to 6!
> >
> > We sincerely hope that our response and detailed revision can also address other reviewers’ concerns. We really appreciate it if they could also re-evaluate the submission. Thanks!

---

> > ### Author Response · Authors · 2021-11-30
> > **Status Update**
> >
> > We appreciate Reviewer XYrN's comments and raising the score to 8!
> >
> > Thanks to Reviewer 16B5 for raising the score to 6! We also appreciate the valuable comments, which helped us significantly improve the paper's strengths.

---

### Decision · Program_Chairs · 2022-01-20

**Decision:**

Accept (Poster)

**Comment:**

Although the initial scores of the paper were not positive, the authors managed to properly address the questions/concerns of the reviewers and the changes they made to the paper convinced the reviewers' to update their scores. This clearly shows that there were flaws in the original presentation of the paper. So, I would recommend the authors to take the reviewers' comments into account when they prepare the camera-ready version of their work.